# Synthesis of New Highly Functionalized 1*H*-Indole-2-carbonitriles via Cross-Coupling Reactions

**DOI:** 10.3390/molecules26175287

**Published:** 2021-08-31

**Authors:** Asma Hrizi, Manon Cailler, Moufida Romdhani-Younes, Yvan Carcenac, Jérôme Thibonnet

**Affiliations:** 1Laboratoire Synthèse et Isolement de Molécules BioActives EA 7502 SIMBA, Department of Chemistry, Faculté des Sciences et Techniques de Tours, Université de Tours, Parc de Grandmont, 37200 Tours, France; asma.hrizi1993@gmail.com (A.H.); manon.cailler@gmail.com (M.C.); yvan.carcenac@univ-tours.fr (Y.C.); 2Department of Chemistry, Faculté des Sciences de Bizerte, Université de Carthage, Zarzouna, Bizerte 7000, Tunisia; moufida.romdhani@gmail.com; 3Laboratoire de Chimie (Bio) Organique, Département de Chimie, Faculté des Sciences de Tunis, Campus Universitaire, Université de Tunis El Manar, Structurale et de Polymères—Synthèse et Études Physico-Chimiques (LR99ES14), Tunis 2092, Tunisia

**Keywords:** 1*H*-indole-2-carbonitriles, Sonogashira, Suzuki–Miyaura, Heck, Stille

## Abstract

An approach for the preparation of polysubstituted indole-2-carbonitriles through a cross-coupling reaction of compounds 1-(but-2-ynyl)-1*H*-indole-2-carbonitriles and 1-benzyl-3-iodo-1*H*-indole-2-carbonitriles is described. The reactivity of indole derivatives with iodine at position 3 was studied using cross-coupling reactions. The Sonogashira, Suzuki–Miyaura, Stille and Heck cross-couplings afforded a variety of di-, tri- and tetra-substituted indole-2-carbonitriles.

## 1. Introduction

Indole skeletons exist as key building blocks in drugs, natural products, pharmaceuticals, alkaloids and agrochemicals and exhibit potent and wide-ranging biological activities [1,2,3,4,5]. The indole scaffold represents probably one of the most important structural subunits for the discovery of new drug candidates [6,7,8,9]. In particular, the derivatives of 2-cyanoindoles gained considerable attention in recent years because of their great importance in biological sciences, and they are also of interest thanks to this nitrile function [10,11,12]. The 2-cyanoindole unit is an example of structural motif building blocks and effective precursors for the synthesis of various indole-fused polycycles [13,14,15,16,17,18,19], substituted 2-cyanoindoles [20,21,22,23,24], addition to nitriles [25,26] and indole heterocycle substitution [27,28]. These compounds exhibit a wide range of biological activities (Figure 1). They are widely used in medicinal chemistry and pharmacological research as antagonist molecules. For example, adrenergic antagonist **A** [14] is a drug that inhibits the function of adrenergic receptors. There are also α-adrenoreceptors that are located on vascular smooth muscle. Antagonists reduce or block the signals of agonists. They can be drugs, which are added to the body for therapeutic reasons, or endogenous ligands. Analog **D** [27] of firefly luciferin is a compound of the class of luciferins, light-emitting chemical compounds. It is found in many species of fireflies (Lampyridae). It is the substrate of luciferase, an enzyme that catalyzes its oxidation into oxyluciferin with concomitant hydrolysis of a molecule of ATP into AMP and PPi accompanied by the emission of a photon of yellow light characteristic of these insects. NMDA receptor antagonists **E** [25] are a class of drugs that work to antagonize or inhibit the action of the *N*-Methyl-D-aspartate receptor (NMDAR). They are commonly used as anesthetics for animals and humans; the state of anesthesia they induce is referred to as dissociative anesthesia. The dopamine D_4_ receptor (D_4_R) **F** [20] plays important roles in cognition, attention and decision making. Novel D_4_R-selective ligands have promise in medication development for neuropsychiatric conditions, including Alzheimer’s disease and substance use disorders. Prostaglandin E2 (PGE2) modulator **G [21]**, subtype (EP2), which is a metabolite of arachidonic acid that binds with and regulates cellular responses to PGE2, is associated with numerous physiological and pathological events in a wide range of tissues. As a stimulatory G protein-coupled receptor, PGE2-induced EP2 activation can activate adenylate cyclase, leading to increased cytoplasmic cAMP levels and activation of protein kinase A. Finally, compound **H [22]** is considered as an antiarrhythmic agent. Additionally, the cyano group is a valuable and readily available functional group for the preparation of various functional groups such as amines, amides, esters, ketones and their carboxyl derivatives [29].

Due to their importance, the development of efficient methodologies for the preparation and functionalization of various cyanoindoles has been the subject of intense research efforts. Direct incorporation of the nitrile function to substituted indoles has been accomplished through a variety of methods. These methods involved various sources of a cyano group including: acetonitrile [30], *tert*-butylisocyanide [31], nitromethane [32], benzyl cyanide [33], Beller’s NCTS (*N*-cyano-*N*-phenyl-*p*-toluenesulfonamide) [34] or Zn(CN)_2_ [35,36].

Palladium-catalyzed cross-coupling reactions are among the most successful transformations in organic synthesis. Thanks to all research work carried out over the years, a large variety of C–C and C–X bond formations and numerous highly active catalytic combinations are currently available [37,38,39,40,41]. The broad interest of this cross-coupling methodology is thus found in many fields of application [42,43]. Driven by our interest in the preparation of substituted 2-cyanoindoles and in conjunction with our successful previous research on palladium cross-coupling reactions, we explored the reactivity of 3-iodo-indole-2-carbonitrile of the residual iodine. This approach allowed the preparation of novel substituted 2-cyanoindoles in position 3 (Figure 1).

The aim of this work was to synthesize new 1*H*-indole-2-carbonitrile derivatives, which could also be useful for drug design. Moreover, the substitution in this type of product in position 3 is important for the development of new molecules with biological interests. The reactivity of iodine in this position was studied using some cross-coupling reactions such as Sonogashira, Suzuki–Miyaura, Stille and Heck reactions, which provided a wide variety of molecules. We report the preparation of 3-iodo-2-carbonitrile derivatives as precursors through the use of the C–I bond in coupling reactions to access a molecular diversity (Figure 1).

## 2. Results and Discussion

### 2.1. Synthesis of 1H-Indole-2-carbonitriles

We began our studies with the reaction of thionyl chloride of **1a**–**d** with a catalytic amount of DMF, which gives access to acyl chloride that reacts with a solution of 25% ammonia to give 1*H*-indole-2-carboxamide derivatives **2a**–**d** in good yields (Scheme 1). The reaction with phosphorus(V) oxychloride on the carboxamide afforded the corresponding 1*H*-indole-2-carbonitriles derivatives **3a**–**d** in good yields (65–85%), following a modified procedure (see Experimental Section) [28].

### 2.2. Synthesis of 1-(3-Phenylprop-2-yn-1-yl)-1H-indole-2-carbonitrile Derivatives

Propargylation reaction (NaH then propargyl bromide in DMF) provides the desired 1-(prop-2-yn-1-yl)-1*H*-indole-2-carbonitrile **4a** in a yield of 75% [10,44]. An efficient method for palladium-catalyzed homocoupling reaction of terminal alkyne in the synthesis of 1*H*-indole-2-carbonitrile derivatives was carried out (Scheme 2). In the presence of 10 mol % of PdCl_2_(PPh_3_)_2_, 10 mol % of CuI and 3 equiv. of Et_3_N, homocoupling of various aromatic iodines gave the corresponding dissymmetric alkynes 1-(3-phenylprop-2-yn-1-yl)-1*H*-indole-2-carbonitrile derivatives **5a**–**f** in moderate to good yields (64–90%) [16]. This Sonogashira cross-coupling reaction was run with various substitutions using electron donor groups and electron-withdrawing groups on the aromatic ring, either in ortho, meta or para.

### 2.3. Synthesis of 1-Benzyl-3-iodo-1H-indole-2-carbonitrile Derivatives

A new series of 2-cyanoindole was also synthesized with an alkynyl substituent in position 3. First, aromatic electrophilic substitution in position 3 by iodine with potassium hydroxide was undertaken (Scheme 3) [45]. The resulting iodine derivatives **6a**–**d** were obtained in yields above 80%. Finally, a protection reaction was carried out with benzyl bromide to give the corresponding 1-benzyl-3-iodo-1*H*-indole-2-carbonitrile derivatives **7a**–**d** in good yields (71–92%) [46].

### 2.4. Sonogashira Reaction on the 1-Benzyl-3-iodo-1H-indole-2-carbonitrile Derivatives

Sonogashira cross-coupling was performed using **7a**–**d** as substrates (Scheme 4) [47,48]. The alkynyl substituent in position 3 has been formed with phenylacetylene derivatives, 10 mol% of palladium (II) and 10 mol% of copper iodide. The reaction of 1-benzyl-3-iodo-1*H*-indole-2-carbonitrile with a variety of aromatic alkynes containing both electron-donating and electron-withdrawing substituents was also examined. Thus, compounds **8a**–**j** were obtained in moderate to good yields (69–90%), and this reaction was carried out with various substituents, either in *ortho* (F, OMe), *meta* (Me) or *para* (Et, F, OMe) and also with pyridine derivatives. These products were purified either by crystallization or by chromatography on silica gel.

Previously, our group reported on the access to iodo-imidazodipyridines from imidazopyridine-2-carbonitriles promoted by Grignard reagent in the presence of iodine and ZnI_2_ through a 6-*endo*-*dig* cyclization [49]. We wished to extend these results to the indole core. Some additional tests of magnesium ethyl bromide on compound **8b** either in diethyl ether or in cyclopentylmethylether (CPME) were made. Changing reaction times and temperatures were carried out. The 6-*endo*-*dig* cyclization product was not isolated, and the addition of Grignard reagent on the cyano group provided the corresponding ketone in a very low yield (4%). However, a test with *n*-BuLi as the organometallic agent yielded the corresponding ketone **8b’** with a full conversion rate, and no cyclization product was observed (Scheme 5). Further efforts to studies on the mechanism and synthetic applications for this type of cyclization are underway in our laboratory.

### 2.5. Suzuki Reaction on the 1-Benzyl-3-iodo-1H-indole-2-carbonitrile Derivatives

Suzuki cross-coupling is one of the most efficient methods for the construction of C–C bonds. Although several other methods are available for this purpose, the Suzuki cross-coupling reaction, which produces biaryls, has proven to be the most popular in recent times. The key advantages of this coupling are the commercial availability of diverse boronic acids that are environmentally safer than other organometallic reagents. Another cross-coupling on nitrile compounds was also performed. The Suzuki reaction was carried out herein with a slight excess of boronic acid using NaHCO_3_ as base and tetrakistriphenylphosphine (10 mol%) as a catalyst in toluene/water mixture as a solvent (Scheme 6) [50,51,52]. Several compounds were synthesized with different *ortho*-, *met*a- or *par*a-aryls substituted by electron donor groups (Me, Et, *t*Bu, OMe) or electron-drawing groups (F, Cl) and with naphthalene derivative. Good yields (79–93%) were also obtained when substituted phenyl groups were used in this cross-coupling reaction (**9a**–**p**). These products were purified either by crystallization or by chromatography on silica gel.

### 2.6. Heck Reaction on the 1-Benzyl-3-iodo-1H-indole-2-carbonitrile Derivatives

Among all palladium-catalyzed cross-couplings, the Mizoroki–Heck reaction allows a direct coupling between heteroatomic compounds and alkenes [53]. This reaction allowed the creation of a sigma bond between two sp^2^-hybridized carbon atoms through a C–H activation. Compared to other cross-coupling processes, this one presents practical and economic advantages such as using simple and readily available materials. Under conditions (DMF, 80 °C, 24 h), coupling of 1-benzyl-3-iodo-1*H*-indole-2-carbonitrile derivatives in the presence of KOAc (6 equiv.), *n*-Bu_4_NCl (2 equiv.), Pd(OAc)_2_ (4 mol%) with diverse olefins afforded a series of cyanoindoles substituted in position 3 (**10a**–**f**) in excellent yields (81–92%) (Scheme 7). The *E*-configuration products were mainly obtained except for the acrylonitrile, for which a mixture of two isomers (*E*/*Z*= 75/25) was observed.

### 2.7. Stille Reaction on the 1-Benzyl-3-iodo-1H-indole-2-carbonitrile Derivatives

Stille coupling was performed using an organotin compound and a catalytic amount of dichlorobis(acetonitrile)palladium(II) in DMF at 40°C (Scheme 8) [54,55]. Under these conditions, the coupling with compound **7a** was successfully achieved, and two examples of coupling products (**11b** and **11c**) were obtained in low to moderate yields (35–40%). No coupling product with tributyl(vinyl)stannane could be obtained (**11a**, 0%). Finally, these compounds were purified by column chromatography with a stationary phase composed of 10% powdered anhydrous K_2_CO_3_ and silica to remove all traces of organotin impurities [56].

## 3. Materials and Methods

### 3.1. General Information

The reagents were purchased from commercial suppliers and used without further purification. Melting points were determined on Büchi B-540 apparatus and are uncorrected. All solvents were dried following the procedure described by Armarego et Chai [57]. ^1^H NMR and ^13^C NMR spectra (from Appendix A) were recorded on a Bruker Avance 300 MHz at 300 and 75 MHz, respectively. ^1^H NMR spectra were recorded in CDCl_3_ or referenced the residual CHCl_3_ at 7.26 ppm (2.50 ppm for DMSO-*d*_6_); ^13^C NMR and J-mod spectra were referenced to the central peak of CDCl_3_ at 77.0 ppm (39.52 ppm for DMSO-*d*_6_). ^19^F NMR was recorded at 282 MHz on the same instrument, using the CFCl_3_ as internal reference (δ 0.0). Chemical shifts were reported in parts per million (ppm, δ), and coupling constants (*J*) were given in Hertz (Hz). Abbreviations for signal coupling are as follows: s, singlet; d, doublet; t, triplet; q, quartet; quin, quintet; sextuplet; sext, dd, doublet of doublets; dq, doublet of quartets; m, multiplet. High-resolution mass spectra (HRMS) were obtained by the electrospray ionization time-of-flight (ESI) mass spectrometry. Thin-layer chromatography (TLC) was performed on TLC silica gel 60 F_254_. Compounds were visualized under UV light (λ = 254 nm) and/or by immersion in a KMnO_4_ solution followed by heating. Products were purified by flash column chromatography on silica gel (0.04–0.063 mm) using various mixtures of EtOAc and petroleum ether (35–60 °C fraction) as eluent. Heating was performed using a magnetic stirrer hotplate and an appropriate-sized heating block. Tributyl(vinyl)stannane was prepared from vinylmagnesium bromide and bis(tributyltin) oxide [58,59]. (*E*)-1-(Tributylstannyl)-2-(trimethylsilyl)ethene was prepared by hydrostannation of (trimethylsilyl)acetylene [60]. (*E*)-Tributyl(3-methylbut-1-en-1-yl)stannane was prepared by the method described by Chong [61]. The compound’s name follows the IUPAC recommendations.

### 3.2. Experimental Section

#### 3.2.1. General Procedure for the Synthesis of Nitriles (Series 3)

To a solution of amide (4.0 g, 25.0 mmol, 1.0 equiv.) in chloroform (75 mL) was added dropwise phosphorus oxychloride (9.24 mL, 99.1 mmol, 4.0 equiv.). The mixture was stirred under reflux for 3 h and then cooled to room temperature. The reaction mixture was quenched by a 25% NH_4_OH aq. Solution (20 mL) and the resulting aqueous layer were extracted with Et_2_O (3 × 30 mL). The organic layers were combined, dried over MgSO_4_ and concentrated under reduced pressure. The crude product was purified by column chromatography using petroleum ether/ethyl acetate (80:20) as eluent.

*1H-Indole-2-carbonitrile* (**3a**): Yellow solid, 79%, 2.83 g. ^1^H NMR (300 MHz, CDCl_3_): *δ* = 8.66 (bs, 1H, N*H*), 7.68 (dq, *J* = 8.1 Hz, 0.9 Hz, 1H, Ar-*H*), 7.45–7.36 (m, 2H, Ar-*H*), 7.25–7.19 (m, 2H, Ar-*H*) ppm. ^13^C NMR (75 MHz, CDCl_3_): *δ* = 137.1 (C_quat_), 126.3 (Ar-*C*H), 126.2 (C_quat_), 122.1 (Ar-*C*H), 121.7 (Ar-*C*H), 114.6 (C_quat_), 114.5 (Ar-*C*H), 112.0 (Ar-*C*H), 106.0 (C_quat_) ppm. These spectroscopic data correspond to the reported data in reference [28].

*5-Methoxy-1H-indole-2-carbonitrile* (**3b**): Yellow solid, 77%, 2.85 g. ^1^H NMR (300 MHz, CDCl_3_): *δ* = 8.57 (bs, 1H, N*H*), 7.31 (dd, *J* = 9.7 Hz, *J* = 0.9 Hz, 1H, Ar-*H*), 7.12 (dd, *J* = 2.1 Hz, *J* = 0.9 Hz, 1H, Ar-*H*), 7.07 (d, *J* = 2.4 Hz, 1H, Ar-H), 7.06–7.03 (m, 1H, Ar-*H*), 6.83 (d, *J* = 1.3 Hz, 1H, Ar-*H*), 3.85 (s, 3H, OC*H*_3_) ppm. ^13^C NMR (75 MHz, CDCl_3_): *δ* = 155.5 (C_quat_), 132.3 (C_quat_), 126.9 (C_quat_), 118.1 (Ar-*C*H), 114.4 (C_quat_), 114.0 (Ar-*C*H), 112.7 (Ar-*C*H), 106.6 (C_quat_), 102.1 (Ar-*C*H), 55.8 (O*C*H_3_) ppm. These spectroscopic data correspond to the reported data in reference [45].

*6-Methoxy-1H-indole-2-carbonitrile* (**3c**): Yellow solid, 77%, 2.85 g. ^1^H NMR (300 MHz, CDCl_3_): *δ* = 8.71 (bs, 1H, N*H*), 7.52 (d, *J* = 8.8 Hz, 1H, Ar-*H*), 7.13 (d, *J* = 1.3 Hz, 1H, Ar-*H*), 6.88 (dd, *J* = 8,8 Hz, 2.1 Hz, 1H, Ar-*H*), 6.83 (d, *J* = 1.3 Hz, 1H, Ar-*H*), 3.86 (s, 3H, OC*H*_3_) ppm. ^13^C NMR (75 MHz, CDCl_3_): *δ* = 159.7 (C_quat_), 138.2 (C_quat_), 122.9 (Ar-*C*H), 120.6 (C_quat_), 114.9 (Ar-CH), 114.8 (C_quat_), 113.4 (Ar-*C*H), 104.8 (C_quat_), 93.8 (*C*H), 55.7 (O*C*H_3_) ppm. These spectroscopic data correspond to the reported data in reference [27].

*5-Fluoro-1H-indole-2-carbonitrile* (**3d**): Yellow solid, 81%, 2.90 g. ^1^H NMR (300 MHz, CDCl_3_): *δ* = 8.64 (bs, 1H, N*H*), 7.37 (dd, *J* = 9.0 Hz, 4.3 Hz, 1H, Ar-*H*), 7.32 (dd, *J* = 8.9 Hz, 2.5 Hz, 1H, Ar-*H*), 7.20–7.12 (m, 2H, Ar-C*H*) ppm. ^13^C NMR (75 MHz, CDCl_3_): *δ* = 158.7 (d, *J* = 238.6 Hz, C_quat_), 133.6 (C_quat_), 126.6 (d, *J* = 10.6 Hz, C_quat_), 115.7 (d, *J* = 27.1 Hz, Ar-*C*H), 114.3 (d, *J* = 5.4 Hz, Ar-*C*H), 114.1 (C_quat_), 113.0 (d, *J* = 9.5 Hz, Ar-*C*H), 107.7 (C_quat_), 106.5 (d, *J* = 23.8 Hz, Ar-*C*H) ppm. ^19^F NMR (282 MHz, CDCl_3_): *δ* = −119.9 ppm. These spectroscopic data correspond to the reported data in reference [45].

#### 3.2.2. Typical Procedure for the Synthesis of Propargyl Compound **4**

In a two-neck round bottom flask, NaH 60% (0.60 g,15.0 mmol, 1.3 equiv.) was dissolved in DMF (15 mL), and indole (7.03 mmol, 1 equiv.) in DMF (10 mL) was added dropwise at 0 °C under argon. The mixture was stirred for 30 min at room temperature. Propargyl bromide 80% (11.4 mmol, 1.3 equiv.) was diluted in DMF and added dropwise into the flask at 0 °C under argon. After 3–4 h of stirring at room temperature, the mixture was hydrolyzed with sat. aq. NH_4_Cl (10 mL), extracted with Et_2_O (7 × 20 mL), and the combined organic layers were washed with brine (6 × 10 mL), dried over MgSO_4_ and evaporated under reduced pressure.

*1-(Prop-2-yn-1-yl)-1H-indole-2-carbonitrile* (**4a**): Yellow solid, 75%, 0.750 g. ^1^H NMR (300 MHz, CDCl_3_): *δ* = 7.69 (dt, *J* = 8.1 Hz, 1H, Ar-*H*), 7.51 (dd, *J* = 8.4 Hz, *J* = 0.9 Hz, 1H, Ar-*H*), 7.48–7.41 (m, 1H, Ar-*H*), 7.29–7.19 (m, 2H, Ar-*H*), 5.04 (s, 2H, C*H*_2_), 2.40 (t, *J* = 2.5 Hz, 1H, C*H*) ppm. ^13^C NMR (75 MHz, CDCl_3_): *δ* = 137.3 (C_quat_), 126.6 (C_quat_), 126.4 (Ar-*C*H), 122.7 (Ar-*C*H), 122.0 (Ar-*C*H), 114.2 (Ar-*C*H), 113.2 (C_quat_), 110.7 (Ar-*C*H), 109.6 (C_quat_), 76.5 (C_quat_), 74.2 (*C*H), 34.4 (*C*H_2_) ppm. These spectroscopic data correspond to the reported data in reference [10].

#### 3.2.3. General Procedure for the Synthesis of Nitriles (Series 5)

A dried Schlenk tube was charged with compound **4a** (1.6 mmol, 1 equiv.), aryl iodide derivative (2.1 mmol, 1.3 equiv.) and DMF (5 mL). The solution was cooled to 0 °C for 15 min under argon ant Et_3_N (4.8 mmol, 3 equiv.) was added into the Schlenk. CuI (0.16 mmol, 10 mol%) and PdCl_2_(PPh_3_)_2_ (0.16 mmol, 10 mol%) were added, and the mixture was stirred overnight at room temperature. The mixture was hydrolyzed with sat. aq. NH_4_Cl (10 mL) and extracted with Et_2_O (7 × 20 mL). The organic layers were washed with brine (2 × 10 mL), dried over MgSO_4_ and evaporated under reduced pressure.

*1-(3-(p-Tolyl)prop-2-yn-1-yl)-1H-indole-2-carbonitrile* (**5a**): Yellow solid, 70%, 252 mg. mp 96–98 °C. ^1^H NMR (300 MHz, CDCl_3_): *δ* = 7.69 (d, *J* = 8.1 *Hz*, 1H, Ar-*H*), 7.59 (dd, *J* = 8.5 Hz, 8.0 Hz, 1H, Ar-*H*), 7.45 (ddd, *J* = 8.4 Hz, *J =* 7.0 Hz, *J =* 1.1 Hz, 1H, Ar-*H*), 7.29 (d, *J =* 8.1 Hz, 2*H*), 7.27–7.21 (m, 2*H*), 7.09 (d, *J =* 7.9 Hz, 2*H*), 5.25 (s, 2H, C*H*_2_), 2.32 (s, 3H, C*H*_3_) ppm. ^13^C NMR (75 MHz, CDCl_3_): *δ* = 139.1 (C_quat_), 137.3 (C_quat_), 131.8 (2Ar-*C*H), 129.1 (2Ar-*C*H), 126.5 (C_quat_), 126.2 (Ar-*C*H), 122.5 (Ar-CH), 121.8 (Ar-*C*H), 118.8 (C_quat_), 113.9 (Ar-*C*H), 113.4 (C_quat_), 110.8 (Ar-*C*H), 109.5 (C_quat_), 85.9 (C_quat_), 81.2 (C_quat_), 35.7 (*C*H_2_), 21.5 (*C*H_3_) ppm. HRMS (ESI): calcd. for C_19_H_15_N_2_ (M + H)^+^ 271.12352; found 271.12347.

*1-(3-(4-Chlorophenyl)prop-2-yn-1-yl)-1H-indole-2-carbonitrile* (**5b**): Yellow solid, 90%, 353 mg). mp 98–100 °C. ^1^H NMR (300 MHz, CDCl_3_): *δ* = 7.69 (d, *J* = 8.1 *Hz*, 1H, Ar-*H*), 7.56 (d, *J* = 8.4 Hz, 1H, Ar-*H*), 7.46 (ddd, *J* = 8.1 Hz, *J* = 7.0 Hz, *J* = 1.0 Hz, 1H, Ar-H), 7.33 (d, *J* = 8.6 Hz, 2H, Ar-*H*), 7.30–7.22 (m, 4H, Ar-*H*), 5.25 (s, 2H, C*H*_2_) ppm. ^13^C NMR (75 MHz, CDCl_3_): *δ* = 137.3 (C_quat_), 135.1 (C_quat_), 133.2 (2Ar-*C*H), 128.8 (2Ar-*C*H), 126.6 (C_quat_), 126.4 (Ar-*C*H), 122.7 (Ar-*C*H), 122.0 (Ar-*C*H), 120.4 (C_quat_), 114.2 (Ar-*C*H), 113.3 (C_quat_), 110.7 (Ar-*C*H), 109.6 (C_quat_), 84.7 (C_quat_), 82.9 (C_quat_), 35.6 (*C*H_2_) ppm. HRMS (ESI): calcd. for C_18_H_12_ClN_2_ (M + H)^+^ 291.06890; found 291.06901. 

*1-(3-(4-(Trifluoromethyl)phenyl)prop-2-yn-1-yl)-1H-indole-2-carbonitrile* (**5c**): Yellow solid, 78%, 341 mg. mp 110–112 °C. ^1^H NMR (300 MHz, CDCl_3_): *δ* = 7.71 (d, *J* = 8.1 *Hz*, 1H, Ar-*H*), 7.60–7.53 (m, 3H, Ar-*H*), 7.53–7.42 (m, 3H, Ar-*H*), 7.29 (m, 2H), 7.29 (dd, *J* = 7.1 Hz, *J* = 0.9 *Hz*, 1H, Ar-*H*), 7.24 (s, 1H, Ar-*H*), 5.28 (s, 2H, C*H*_2_) ppm. ^13^C NMR (75 MHz, CDCl_3_): *δ* = 137.4 (C_quat_), 132.3 (2Ar-*C*H), 130.7 (q, *J* = 32.5 Hz, C_quat_), 126.6 (C_quat_), 125.7 (q, *J* = 1.3 Hz, C_quat_), 125.4 (q, *J* = 3.7 Hz, 2Ar-*C*H), 123.9 (q, *J* = 272.8 Hz, C_quat_), 122.7 (Ar-*C*H), 122.1 (Ar-*C*H), 114.3 (Ar-*C*H), 113.3 (C_quat_), 110.6 (Ar-*C*H), 109.6 (C_quat_), 84.4 (C_quat_), 84.3 (C_quat_), 35.6 (*C*H_2_) ppm. HRMS (ESI): calcd. for C_19_H_12_F_3_N_2_ (M + H)^+^ 325.09471; found 325.09395.

*1-(3-(2-Cyanophenyl)prop-2-yn-1-yl)-1H-indole-2-carbonitrile* (**5d**): Yellow solid, 64%, 379 mg. mp 137–139 °C. ^1^H NMR (300 MHz, CDCl_3_): *δ* = 7.71–7.62 (m, 2H, Ar-*H*), 7.54–7.47 (m, 3H, Ar-*H*), 7.46–7.39 (m, 2H, Ar-*H*), 7.29–7.22 (m, 2H, Ar-*H*), 5.34 (s, 2H, CH_2_) ppm. ^13^C NMR (75 MHz, CDCl_3_): *δ* = 137.4 (C_quat_), 132.9 (Ar-*C*H), 132.7 (Ar-*C*H), 132.5 (Ar-*C*H), 129.2 (Ar-*C*H), 126.63 (Ar-*C*H), 126.6 (C_quat_), 125.8 (C_quat_), 122.6 (Ar-*C*H), 122.1 (Ar-*C*H), 117.3 (C_quat_), 115.7 (C_quat_), 114.4 (Ar-*C*H), 113.3 (C_quat_), 111.0 (Ar-*C*H), 109.5 (C_quat_), 88.3 (C_quat_), 81.9 (C_quat_), 35.6 (*C*H_2_) ppm. HRMS (ESI): calcd. for C_19_H_11_N_3_ (M + H)^+^ 282.10257; found 282.10179.

*1-(3-(2-Methoxyphenyl)prop-2-yn-1-yl)-1H-indole-2-carbonitrile* (**5e**): Yellow solid, 71%, 269 mg. mp 96–98 °C. ^1^H NMR (300 MHz, CDCl_3_): *δ* = 7.70–7.64 (m, 2H, Ar-*H*), 7.45 (ddd, *J* = 8.5 Hz, *J* = 7.1 Hz, *J* = 1.1 Hz, 1H, Ar-*H*), 7.35 (dd, *J* = 7.5 Hz, *J* = 1.6 Hz, 1H, Ar-*H*), 7.32–7.19 (m, 3H, 3Ar-*H*), 6.87 (td, *J* = 7.5 Hz, *J* = 0.9 Hz, 1H, Ar-*H*), 6.84 (d, *J* = 8.4 Hz, Ar-*H*), 5.29 (s, 2H, C*H*_2_), 3.84 (s, OC*H*_3_) ppm. ^13^C NMR (75 MHz, CDCl_3_): *δ* = 160.4 (C_quat_), 137.4 (C_quat_), 133.9 (Ar-*C*H), 130.4 (Ar-*C*H), 126.6 (C_quat_), 126.1 (Ar-*C*H), 122.5 (Ar-*C*H), 121.8 (Ar-*C*H), 120.5 (Ar-*C*H), 113.9 (Ar-*C*H), 113.4 (C_quat_), 111.2 (2Ar-*C*H + C_quat_), 110.8 (Ar-*C*H), 109.6 (C_quat_), 85.8 (C_quat_), 82.4 (C_quat_), 55.8 (O*C*H_3_), 36.1 (*C*H_2_) ppm. HRMS (ESI): calcd. for C_19_H_15_N_2_O (M + H)^+^ 287.11844; found 287.11835.

*1-(3-(3,4-Dichlorophenyl)prop-2-yn-1-yl)-1H-indole-2-carbonitrile* (**5f**): Yellow solid, 87%, 381 mg. mp 115–117 °C. ^1^H NMR (300 MHz, CDCl_3_): *δ* = 7.70 (d, *J* = 8.1 Hz, 1H, Ar-*H*), 7.54 (d, *J* = 8.4 Hz, 1H, Ar-*H*), 7.50–7.44 (m, 2H, Ar-*H*), 7.36 (d, *J =* 8.3 Hz, 1H, Ar-*H*), 7.27 (dd, *J =* 7.0 Hz, *J* = 1.0 Hz, 1H, Ar-*H*), 7.23 (s, 1H, Ar-*H*), 7.20 (dd, *J* = 8.4 Hz, *J* = 1.9 Hz, 1H, Ar-*H*), 5.25 (s, 2H, C*H*_2_) ppm. ^13^C NMR (75 MHz, CDCl_3_): *δ* = 137.3 (C_quat_), 133.5 (Ar-*C*H), 132.7 (C_quat_), 131.1 (Ar-*C*H), 130.5 (Ar-*C*H), 126.6 (C_quat_), 126.5 (Ar-*C*H), 122.7 (Ar-*C*H), 122.1 (Ar-*C*H), 121.8 (C_quat_), 114.3 (Ar-*C*H), 113.3 (C_quat_), 111.8 (C_quat_), 110.5 (Ar-*C*H), 109.6 (C_quat_), 83.9 (C_quat_), 83.5 (C_quat_), 35.5 (*C*H_2_) ppm. HRMS (ESI): calcd. for C_18_H_10_^35^Cl_2_N_2_ (M + H)^+^ 325.02938; found 325.02841.

#### 3.2.4. General Procedure for the Synthesis of Nitriles (Series 6)

To a solution of 1*H*-indole-2-carbonitrile (2.0 g, 14.1 mmol, 1.0 equiv.) in DMF (10 mL), KOH (0.79 g, 50.3 mmol, 3.6 equiv.) was added in small portions. The mixture was stirred for 30 min at room temperature. Then, a solution of iodine (3.57 g, 14.1 mmol, 1.0 equiv.) in DMF (3 mL) was added dropwise at 0 °C, and the mixture was stirred for 4 h at room temperature. The mixture was then poured into a mixture of water (600 mL) and sat. aq. NH_4_Cl (40 mL) and stirred for 30 min. The precipitate was filtered on a Büchner funnel and dried under vacuum for 2 h.

*3-Iodo-1H-indole-2-carbonitrile* (**6a**): White solid, 77%, 2.90 g. ^1^H NMR (300 MHz, CDCl_3_): *δ* = 8.98 (bs, 1H, N*H*), 7.49 (dq, *J* = 8.1 Hz, *J* = 0.9 Hz, 1H, Ar-*H*), 7.47–7.39 (m, 2H, Ar-*H*), 7.30 (ddd, *J* = 8.1 Hz, *J* = 6.2 Hz, *J* = 1.8 Hz, 1H, Ar-*H*) ppm. ^13^C NMR (75 MHz, DMSO-d_6_): *δ* = 137.0 (C_quat_), 128.9 (C_quat_), 126.6 (Ar-*C*H), 122.0 (Ar-*C*H), 121.8 (Ar-*C*H), 114.3 (C_quat_), 112.8 (Ar-*C*H), 111.4 (C_quat_), 72.0 (C_quat_) ppm. These spectroscopic data correspond to the reported data in reference [46].

*3-Iodo-5-methoxy-1H-indole-2-carbonitrile* (**6b**): White solid, 78%, 2.70 g. ^1^H NMR (300 MHz, DMSO-d_6_): *δ* = 12.74 (bs, 1H, N*H*), 7.40 (dd, *J* = 9.0 Hz, *J* = 0.4 Hz, 1H, Ar-*H*), 7.05 (dd, *J* = 9.0 Hz, 2.5 Hz, 1H, Ar-H), 6.77 (d, *J =* 2.3 Hz, 1H, Ar-*H*), 3.82 (s, 3H, OC*H*_3_) ppm. ^13^C NMR (75 MHz, DMSO-d_6_): *δ* = 155.4 (C_quat_), 132.1 (C_quat_), 129.3 (C_quat_), 118.3 (Ar-*C*H), 114.4 (C_quat_), 114.0 (Ar-*C*H), 111.2 (C_quat_), 101.3 (Ar-*C*H), 70.8 (C_quat_), 55.4 (O*C*H_3_) ppm. HRMS (ESI): calcd. for C_10_H_8_IN_2_O (M + H)^+^ 298.96813; found 298.96711.

*3-Iodo-6-methoxy-1H-indole-2-carbonitrile* (**6c**): White solid, 81%, 2.80 g. ^1^H NMR (300 MHz, CDCl_3_): *δ* = 9.09 (bs, 1H, N*H*), 7.31 (d, *J* = 9.2 Hz, 1H, Ar-*H*), 6.92 (dd, *J* = 9.2 Hz, 2.1 Hz, 1H, Ar-*H*), 6.80 (d, *J* = 2.1 Hz, 1H, Ar-H), 3.87 (s, 3H, OCH_3_) ppm. ^13^C NMR (75 MHz, CDCl_3_): *δ* = 160.6 (C_quat_), 137.8 (C_quat_), 124.0 (C_quat_), 123.6 (Ar-*C*H), 114.4 (Ar-*C*H), 114.4 (C_quat_), 110.4 (C_quat_), 93.8 (Ar-*C*H), 72.5 (C_quat_), 55.8 (O*C*H_3_) ppm. HRMS (ESI): calcd. for C_10_H_8_IN_2_O (M + H)^+^ 298.96813; found 298.96741.

*5-Fluoro-3-iodo-1H-indole-2-carbonitrile* (**6d**): White solid, 73%, 2.60 g. ^1^H NMR (300 MHz, DMSO-d_6_): *δ* = 12.05 (bs, 1H, N*H*), 8.03 (ddd, *J* = 9.0 Hz, 4.3 Hz, 0.4 Hz, 1H, Ar-*H*), 7.69 (td, *J* = 9.2 Hz, *J* = 2.5 Hz, 1H, Ar-*H*), 7.58 (ddd, *J* = 9.0 Hz, 2.5 Hz, 0.4 Hz, 1H, Ar-*H*) ppm. ^13^C NMR (75 MHz, Acetone-d_6_): *δ* = 159.9 (d, *J* =238.1 Hz, C_quat_), 134.6 (C_quat_), 130.6 (d, *J* = 10.7 Hz, C_quat_), 116.6 (d, *J* = 27.2 Hz, Ar–*C*H), 115.3 (d, *J* = 9.5 Hz, Ar–*C*H), 114.7 (C_quat_), 114.0 (C_quat_), 107.1 (d, *J* = 24.9 Hz, Ar–*C*H), 70.1 (d, *J* = 5.6 Hz, C_quat_) ppm. ^19^F NMR (282 MHz, CDCl_3_): *δ* = −119.6 Hz. HRMS (ESI): calcd. for C_9_H_5_FIN_2_ (M + H) ^+^ 286.94814; found 286.20350.

#### 3.2.5. General Procedure for the Synthesis of Nitriles (Series 7)

To compound **6a** (2.0 g, 7.4 mmol, 1 equiv.) in DMF (10 mL), NaH (0.17 g, 44 mmol, 1.2 equiv.) was added portion-wise at 0 °C under argon. The mixture was stirred for 30 min at room temperature. Then, benzyl bromide (0.57 mL, 0.0048 mol, 1.3 equiv.) was added dropwise at 0 °C. The mixture was stirred at room temperature for 3–4 h and hydrolyzed with sat. aq. NH_4_Cl (10 mL). The aqueous phase was extracted with DCM (3 × 20 mL), and the combined organic layers were dried over MgSO_4_ and concentrated under reduced pressure. The product was purified by column chromatography using petroleum ether/ethyl acetate (80:20) as eluent.

*1-Benzyl-3-iodo-1H-indole-2-carbonitrile* (**7a**): White solid, 71%, 1.90 g. mp 144–146 °C. ^1^H NMR (300 MHz, CDCl_3_): *δ* = 7.50 (dt, *J* = 8.1 Hz, 1.0 Hz, 1H, Ar-*H*), 7.41–7.38 (m, 1H, Ar-H), 7.37–7.27 (m, 5H, Ar-*H*), 7.18 (dd, *J* = 7.5 Hz, *J* = 2.0 Hz, 2H, Ar-*H*), 5.51 (s, 2H, C*H*_2_) ppm. ^13^C NMR (75 MHz, CDCl_3_): *δ* = 137.4 (C_quat_), 135.6 (C_quat_), 129.8 (C_quat_), 129.2 (2Ar-*C*H), 128.5 (Ar-*C*H), 127.2 (Ar-*C*H), 127.0 (2Ar-*C*H), 123.2 (Ar-*C*H), 122.6 (Ar-*C*H), 115.4 (C_quat_), 113.4 (C_quat_), 111.1 (Ar-*C*H), 70.6 (C_quat_), 50.1 (*C*H_2_) ppm. HRMS (ESI): calcd. for C_16_H_12_IN_2_ (M + H)^+^ 359.00397; found 359.00293.

*1-Benzyl-3-iodo-5-methoxy-1H-indole-2-carbonitrile* (**7b**): White solid, 81%, 2.10 g. ^1^H NMR (300 MHz, CDCl_3_): *δ* = 7.35–7.27 (m, 3H, Ar-*H*), 7.20 (d, *J* = 9.1 Hz, 1H, Ar-*H*), 7.15 (dd, *J* = 7.5 Hz, *J* = 2.4 Hz, 2H, Ar-*H*), 7.04 (dd, *J* = 9.1 Hz, *J* = 2.4 Hz*,* 1H, Ar-*H*), 6.82 (d, *J* = 2.4 Hz, 1H, Ar-*H*), 5.47 (s, 2H, CH_2_), 3.88 (s, 3H, OCH_3_) ppm. ^13^C NMR (75 MHz, CDCl_3_): *δ* = 156.3 (C_quat_), 135.7 (C_quat_), 132.7 (C_quat_), 130.2 (C_quat_), 129.2 (2Ar-*C*H), 128.4 (Ar-*C*H), 126.9 (2Ar-*C*H), 119.1 (Ar-*C*H), 115.3 (C_quat_), 113.5 (C_quat_), 112.2 (Ar-*C*H), 102.7 (Ar-*C*H), 69.4 (C_quat_), 55.9 (O*C*H_3_), 50.2 (*C*H_2_) ppm. HRMS (ESI): calcd. for C_17_H_14_IN_2_O (M + H)^+^ 389.01453; found 389.01495.

*1-Benzyl-3-iodo-6-methoxy-1H-indole-2-carbonitrile* (**7c**): White solid, 92%, 2.40 g. ^1^H NMR (300 MHz, CDCl_3_): *δ* = 7.38–7.28 (m, 4H, Ar-H), 7.17 (dd, *J* = 7.6 Hz, *J* = 2.1 Hz, 2H, Ar-*H*), 6.92 (dd, *J* = 8.9 Hz, *J* = 2.1 Hz, 1H, Ar-*H*), 6.66 (d, *J* = 2.1 Hz, 1H, Ar-*H*), 5.43 (s, 2H, C*H*_2_), 3.81 (s, 3H, OC*H*_3_) ppm. ^13^C NMR (75 MHz, CDCl_3_): *δ* = 160.4 (C_quat_), 138.5 (C_quat_), 135.6 (C_quat_), 129.2 (Ar-*C*H), 128.4 (Ar-*C*H), 127.0 (Ar-*C*H), 124.2 (C_quat_), 124.0 (Ar-*C*H), 114.2 (C_quat_), 113.9 (Ar-*C*H), 113.8 (C_quat_), 92.9 (Ar-*C*H), 70.9 (C_quat_), 55.8 (O*C*H_3_), 50.0 (*C*H_2_) ppm. HRMS (ESI): calcd. for C_17_H_14_IN_2_O (M + H)^+^ 389.01508; found 389.01492.

*1-Benzyl-5-fluoro-3-iodo-1H-indole-2-carbonitrile* (**7d**): White solid, 76%, 2.0 g. ^1^H NMR (300 MHz, CDCl_3_): *δ* = 7.37–7.27 (m, 4H, Ar-*H*), 7.26–7.23 (m, 1H, Ar-*H*), 7.19–7.11 (m, 3H, Ar-*H*), 5.49 (s, 2H, C*H*_2_) ppm. ^13^C NMR (75 MHz, CDCl_3_): *δ* = 159.3 (d, *J* = 241.0 Hz, C_quat_), 135.3 (C_quat_), 134.0 (C_quat_), 130.3 (d, *J* = 10.6 Hz, C_quat_), 129.3 (2Ar–*C*H), 128.6 (Ar–*C*H), 126.9 (2Ar–*C*H), 116.9 (C_quat_), 116.5 (d, *J* = 27.1 Hz, Ar–*C*H), 113.0 (C_quat_), 112.4 (d, *J* = 9.4 Hz, Ar–*C*H), 107.8 (d, *J* = 24.7 Hz, Ar–*C*H), 69.4 (d, *J* = 5.6 Hz, C_quat_), 50.4 (*C*H_2_) ppm. ^19^F (282 MHz, CDCl_3_): *δ* = −119.9 ppm. HRMS (ESI): calcd. for C_16_H_11_FIN_2_ (M + H)^+^ 376.99509; found 376.99497.

#### 3.2.6. General Procedure for Sonogashira Coupling (Series 8)

A Schlenk tube was charged with phenylacetylene (1.08 mmol, 1.3 equiv.), Et_3_N (0.34 mL, 2.52 mmol, 3.0 equiv.), DMF (3 mL), PdCl_2_(PPh_3_)_2_ (58 mg, 0.083 mmol, 10 mol%), compound **7a** (300 mg, 0.837 mmol, 1 equiv.) and CuI (160 mg, 0.084 mmol, 10 mol%). The tube was evacuated and backfilled with argon, and the mixture was stirred at room temperature overnight. Then, the mixture was hydrolyzed with sat. aq. NH_4_Cl (10 mL), extracted by Et_2_O (3 × 20 mL), and the combined organic layers were washed with brine (6 × 10 mL), dried over MgSO_4_ and concentrated under reduced pressure. The product was purified by column chromatography using petroleum ether/ethyl acetate (80:20) as eluent.

*1-Benzyl-3-(phenylethynyl)-1H-indole-2-carbonitrile* (**8a**): Yellow solid, 70%, 195 mg. mp 125–127 °C. ^1^H NMR (300 MHz, CDCl_3_): *δ* = 7.88 (dt, *J* = 8.0 Hz, *J* = 1.0 Hz, 1H, Ar-*H*), 7.67–7.62 (m, 2H, Ar-*H*), 7.44–7.39 (m, 4H, Ar-*H*), 7.36–7.28 (m, 5H, Ar-*H*), 7.20 (dd, *J* = 7.6 Hz, 1.9 Hz, 2H, Ar-*H*), 5.48 (s, 2H, CH_2_) ppm. ^13^C NMR (75 MHz, CDCl_3_): *δ* = 137.1 (C_quat_), 135.6 (C_quat_), 131.9 (2Ar-*C*H), 129.2 (2Ar-*C*H), 128.8 (Ar-*C*H), 128.6 (2Ar-*C*H), 128.4 (Ar-*C*H), 127.5 (C_quat_), 127.0 (Ar-*C*H), 127.0 (2Ar-*C*H), 123.0 (C_quat_), 122.4 (Ar-*C*H), 121.8 (Ar-*C*H), 112.8 (2C_quat_), 111.1 (Ar-*C*H), 109.8 (C_quat_), 97.2 (C_quat_), 79.6 (C_quat_), 49.6 (*C*H_2_) ppm. HRMS (ESI): calcd. for C_24_H_17_N_2_ (M + H)^+^ 333.13863; found 333.13760.

*1-Benzyl-3-(p-tolylethynyl)-1H-indole-2-carbonitrile* (**8b**): Yellow solid, 81%, 235 mg. mp 163–165°C. ^1^H NMR (300 MHz, CDCl_3_): *δ* = 7.86 (d, *J* = 8.2 Hz, 1H, Ar-*H*), 7.51 (d, *J* = 8.1 Hz, 2H, Ar-*H*), 7.43–7.26 (m, 6H, Ar-*H*), 7.22–7.16 (m, 3H, Ar-*H*), 5.48 (s, 2H, CH_2_), 2.40 (s, 3H, CH_3_) ppm. ^13^C NMR (75 MHz, CDCl_3_): *δ* = 139.0 (C_quat_), 137.1 (C_quat_), 135.6 (C_quat_), 131.7 (Ar-*C*H), 129.3 (Ar-*C*H), 129.1 (Ar-*C*H), 128.4 (Ar-*C*H), 127.5 (C_quat_), 127.0 (Ar-*C*H), 122.3 (Ar-*C*H), 121.8 (Ar-*C*H), 119.9 (C_quat_), 112.9 (C_quat_), 112.6 (C_quat_), 111.0 (Ar-*C*H), 110.1 (C_quat_), 97.4 (C_quat_), 79.0 (C_quat_), 49.5 (*C*H_2_), 21.7 (*C*H_3_) ppm. HRMS (ESI): calcd. for C_25_H_19_N_2_ (M + H)^+^ 347.15482; found 346.15489.

*1-Benzyl-3-((4-fluorophenyl)ethynyl)-1H-indole-2-carbonitrile* (**8c**): Brown solid, 69%, 202 mg. mp 180–182 °C. ^1^H NMR (300 MHz, CDCl_3_): *δ* = 7.85 (dt, *J* = 8.0 Hz, *J* = 1.0 Hz, 1H, Ar-*H*), 7.60 (dd, *J* = 8.9 Hz, *J* = 5.4 Hz, 2H, Ar-*H*), 7.40 (ddd, *J* = 8.4 Hz, *J* = 6.6 Hz, *J* = 1.2 Hz, 1H, Ar-*H*), 7.35–7.27 (m, 5H, Ar-*H*), 7.21–7.18 (m, 2H, Ar-*H*), 7.08 (t, *J* = 8.7 Hz, 2H, Ar-*H*), 5.48 (s, 2H, CH_2_) ppm. ^13^C NMR (75 MHz, CDCl_3_): *δ* = 162.9 (d, *J* = 250.3 Hz, C_quat_), 137.1 (C_quat_), 135.6 (C_quat_), 133.8 (d, *J* = 8.4 Hz, Ar-*C*H), 129.2 (2Ar–*C*H), 128.5 (Ar-*C*H), 127.5 (C_quat_), 127.1 (Ar-*C*H), 127.0 (2Ar-*C*H), 122.4 (Ar-*C*H), 121.7 (Ar-*C*H), 119.1 (d*, J* = 3.5 Hz, C_quat_), 115.9 (d, *J* = 22.1 Hz, Ar-*C*H), 112.9 (C_quat_), 112.8 (C_quat_), 111.1 (Ar-*C*H), 109.7 (C_quat_), 96.0 (C_quat_), 79.4 (C_quat_), 49.6 (*C*H_2_) ppm. ^19^F NMR (282 MHz, CDCl_3_): *δ* = −110.2 ppm. HRMS (ESI): calcd. for C_24_H_16_FN_2_ (M-H)^+^ 351.12920; found 351.12816.

*1-Benzyl-3-((4-methoxyphenyl)ethynyl)-1H-indole-2-carbonitrile* (**8d**): Brown solid, 83%, 252 mg. mp 137–139 °C. ^1^H NMR (300 MHz, CDCl_3_): *δ* = 7.85 (dt, *J* = 8.0 Hz, *J* = 1.0 Hz, 1H, Ar-*H*), 7.56 (d, *J* = 8.9 Hz, 2H, Ar-*H*), 7.40 (ddd, *J* = 8.4 Hz, *J* = 6.7 Hz, *J* = 1.2 Hz, 1H, Ar-*H*), 7.36–7.24 (m, 6H, Ar-H), 7.19 (dd, *J* = 7.8 Hz, *J* = 2.0 Hz, 1H, Ar-H), 6.91 (d, *J* = 8.9 Hz, 2H, Ar-*H*), 5.47 (s, 2H, C*H*_2_), 3.85 (s, 3H, OC*H*_3_) ppm. ^13^C NMR (75 MHz, CDCl_3_): *δ* = 160.0 (C_quat_), 137.0 (C_quat_), 135.6 (C_quat_), 133.3 (2Ar-*C*H), 129.1 (2Ar-*C*H), 128.3 (Ar-*C*H), 127.4 (C_quat_), 126.9 (2Ar-*C*H), 122.2 (Ar-*C*H), 121.7 (Ar-*C*H), 115.0 (C_quat_), 114.2 (Ar-*C*H), 112.9 (C_quat_), 112.4 (C_quat_), 111.0 (Ar-*C*H), 110.2 (C_quat_), 97.2 (C_quat_), 78.3 (C_quat_), 55.4 (O*C*H_3_), 49.4 (*C*H_2_) ppm. HRMS (ESI): calcd. for C_25_H_19_N_2_O (M + H)^+^ 363.14919; found 363.14962.

*1-Benzyl-3-((4-ethylphenyl)ethynyl)-1H-indole-2-carbonitrile* (**8e**): Brown solid, 83%, 251 mg. mp 94–96 °C.^1^H NMR (300 MHz, CDCl_3_): *δ* = 7.86 (dt, *J* = 8.0 Hz, *J* = 1.0 Hz, 1H, Ar-*H*), 7.54 (d, *J* = 8.3 Hz, 2H, Ar-*H*), 7.40 (ddd, *J* = 8.4 Hz, *J* = 6.6 Hz, *J* = 1.2 Hz, 1H, Ar-*H*), 7.36–7.28 (m, 5H, Ar-*H*), 7.24–7.18 (m, 4H, Ar-*H*), 5.48 (s, 2H, C*H*_2_), 2.69 (q, *J* = 7.6 Hz, 2H, C*H*_2_), 1.26 (t, *J* = 7.6 Hz, 3H, C*H*_3_) ppm. ^13^C NMR (75 MHz, CDCl_3_): *δ* = 145.3 (C_quat_), 137.0 (C_quat_), 135.6 (C_quat_), 131.8 (2Ar-*C*H), 129.1 (2Ar-*C*H), 128.4 (Ar-*C*H), 128.1 (2Ar-*C*H), 127.5 (C_quat_), 127.0 (3Ar-*C*H), 122.3 (Ar-*C*H), 121.8 (Ar-*C*H), 120.1 (C_quat_), 112.9 (C_quat_), 112.6 (C_quat_),111.0 (Ar-*C*H), 110.1 (C_quat_), 97.4 (C_quat_), 79.0 (C_quat_), 49.5 (*C*H_2_), 29.0 (*C*H_2_), 15.5 (*C*H_3_) ppm. HRMS (ESI): calcd. for C_26_H_21_N_2_ (M + H)^+^ 361.16993; found 361.16903.

*1-Benzyl-3-(pyridin-3-yl-ethynyl)-1H-indole-2-carbonitrile* (**8f**): Brown solid, 78%, 216 mg. mp 138–140 °C. ^1^H NMR (300 MHz, CDCl_3_): *δ* = 8.67 (ddd, *J* = 4.9 Hz, *J* = 1.8 Hz, *J* = 1.0 Hz, 1H, Ar-*H*), 7.93 (dt, *J* = 8.1 Hz, *J* = 1.0 Hz, 1H, Ar-*H*), 7.72 (dt, *J* = 7.8 Hz, *J* = 1.8 Hz, 1H, Ar-*H*), 7.63 (dt, *J* = 7.8 Hz, *J* = 1.2 Hz, 1H, Ar-*H*), 7.41 (ddd, *J* = 8.4 Hz, *J* = 6.6 Hz, *J* = 1.2 Hz, 1H, Ar-*H*), 7.37–7.26 (m, 6H, Ar-*H*), 7.19 (dd, *J* = 7.7 Hz, *J* = 2.1 Hz, 2H, Ar-*H*), 5.50 (s, 2H, C*H*_2_) ppm. ^13^C NMR (75 MHz, CDCl_3_): *δ* = 150.1 (Ar-*C*H), 143.0 (C_quat_), 136.9 (C_quat_), 136.2 (Ar-*C*H), 135.3 (C_quat_), 129.1 (2Ar-*C*H), 128.3 (Ar-*C*H), 127.6 (Ar-*C*H), 127.4 (C_quat_), 127.0 (Ar-*C*H), 126.8 (2Ar-*C*H), 123.0 (Ar-*C*H), 122.5 (Ar-*C*H), 121.8 (Ar-*C*H), 113.5 (C_quat_), 112.4 (C_quat_), 111.0 (Ar-*C*H), 108.5 (C_quat_), 95.9 (C_quat_), 79.5 (C_quat_), 49.5 (*C*H_2_) ppm. HRMS (ESI): calcd. for C_23_H_16_N_3_ (M + H)^+^ 334.13387; found 334.13301.

*1-Benzyl-3-((2-fluorophenyl)ethynyl)-1H-indole-2-carbonitrile* (**8g**): Brown solid, 76%, 222 mg. mp 178–180 °C. ^1^H NMR (300 MHz, CDCl_3_): *δ* = 7.88 (dt, *J* = 8.0 Hz, *J* = 0.9 Hz, 1H, Ar-*H*), 7.60 (td, *J* = 7.2 Hz, *J* = 1.9 Hz, 1H, Ar-*H*), 7.41 (ddd, *J* = 8.4 Hz, *J* = 6.7 Hz, *J* = 1.2 Hz, 1H, Ar-*H*), 7.38–7.27 (m, 6H, Ar-H), 7.21–7.11 (m, 4H, Ar-H), 5.50 (s, 2H, CH_2_) ppm. ^13^C NMR (75 MHz, CDCl_3_): *δ* = 162.7 (d, *J* = 252.3 Hz, C_quat_), 137.0 (C_quat_), 135.5 (C_quat_), 133.5 (Ar-*C*H), 130.5 (d, *J* = 7.9 Hz, Ar-*C*H), 129.1 (2Ar-*C*H), 128.4 (Ar-*C*H), 127.5 (C_quat_), 127.1 (Ar-*C*H), 127.0 (2Ar-*C*H), 124.1 (d, *J* = 3.7 Hz, Ar-*C*H), 122.6 (Ar-*C*H), 121.8 (Ar-*C*H), 115.7 (d*, J* = 20.7 Hz, Ar-*C*H), 112.8 (C_quat_), 112.7 (C_quat_), 111.6 (d, *J* = 15.7 Hz, C_quat_), 111.1 (Ar-*C*H), 109.3 (C_quat_), 90.4 (C_quat_), 84.7 (C_quat_), 49.5 (*C*H_2_) ppm. ^19^F NMR (282 MHz, CDCl_3_): *δ* = −112.1 ppm. HRMS (ESI): calcd. for C_24_H_16_FN_2_ (M + H)^+^ 351.12920; found 351.12956.

*1-Benzyl-5-methoxy-3-((4-methoxyphenyl)ethynyl)-1H-indole-2-carbonitrile* (**8h**): Brown solid, 90%, 272 mg. ^1^H NMR (300 MHz, CDCl_3_): *δ* = 7.57 (d, *J* = 8.9 Hz, 2H, Ar-*H*), 7.35–7.28 (m, 3H, Ar-*H*), 7.20 (d, *J* = 9.1 Hz, 1H, Ar-*H*), 7.16 (dd, *J* = 7.4 Hz, *J* = 2.4 Hz, 2H, Ar-*H*), 7.04 (dd, *J* = 9.1 Hz, *J* 2.5 Hz, 1H, Ar-*H*), 6.91 (d, *J* = 8.9 Hz, 2H, Ar-*H*), 5.44 (s, 2H, C*H*_2_), 3.89 (s, 3H, OC*H*_3_), 3.85 (s, 3H, OC*H*_3_) ppm. ^13^C NMR (75 MHz, CDCl_3_): *δ* = 160.0 (C_quat_), 156.1 (C_quat_), 135.7 (C_quat_), 133.4 (2Ar-C*H*), 132.4 (C_quat_), 129.1 (2Ar-*C*H), 128.4 (Ar-*C*H), 128.1 (C_quat_), 126.9 (2Ar-CH), 118.6 (Ar-*C*H), 115.1 (C_quat_), 114.2 (2Ar-*C*H), 113.0 (C_quat_), 112.6 (C_quat_), 112.1 (Ar-*C*H), 109.3 (C_quat_), 101.6 (Ar-*C*H), 97.0 (C_quat_), 78.4 (C_quat_), 55.9 (O*C*H_3_), 55.5 (O*C*H_3_), 49.7 (*C*H_2_) ppm. HRMS (ESI): calcd. for C_26_H_21_N_2_O_2_ (M + H)^+^ 393.15975; found 393.16029.

*1-Benzyl-6-methoxy-3-((2-methoxyphenyl)ethynyl)-1H-indole-2-carbonitrile* (**8i**): Brown solid, 83%, 252 mg. ^1^H NMR (300 MHz, CDCl_3_): *δ* = 7.74 (d, *J* = 8.8 Hz, 1H, Ar-*H*), 7.56 (dd, *J* = 7.6 Hz, *J* = 1.6 Hz, 1H, Ar-*H*), 7.39–7.28 (m, 4H, Ar-*H*), 7.17 (dd, *J* = 7.6 Hz, *J* = 1.6 Hz, 2H, Ar-*H*), 6.98 (dd, *J* = 7.6 Hz, J = 1.0 Hz, 1H, Ar-*H*), 6.96–6.90 (m, 2H, Ar-*H*), 6.67 (d, *J* = 2.1 Hz, 1H, Ar-*H*), 5.41 (s, 2H, C*H*_2_), 3.96 (s, 3H, OC*H*_3_), 3.80 (s, 3H, OC*H*_3_) ppm. ^13^C NMR (75 MHz, CDCl_3_): *δ* = 160.3 (C_quat_), 160.1 (C_quat_), 138.2 (C_quat_), 135.7 (C_quat_), 133.5 (Ar-*C*H), 130.2 (Ar-*C*H), 129.1 (2Ar-*C*H), 128.3 (Ar-*C*H), 126.9 (2Ar-*C*H), 122.7 (Ar-*C*H), 121.8 (C_quat_), 120.6 (Ar-*C*H), 113.4 (Ar-*C*H), 113.2 (C_quat_), 112.4 (C_quat_), 111.4 (C_quat_), 111.0 (Ar-*C*H), 110.5 (C_quat_), 93.5 (C_quat_), 93.2 (Ar-*C*H), 83.8 (C_quat_), 56.0 (O*C*H_3_), 55.7 (O*C*H_3_), 49.4 (*C*H_2_) ppm. HRMS (ESI): calcd. for C_26_H_21_N_2_O_2_ (M + H)^+^ 393.15975; found 393.15951.

*1-Benzyl-5-fluoro-3-(m-tolylethynyl)-1H-indole-2-carbonitrile* (**8j**): Brown solid, 86%, 250 mg. ^1^H NMR (300 MHz, CDCl_3_): *δ* = 7.55–7.46 (m, 3H), 7.38–7.27 (m, 4H), 7.25–7.08 (m, 5H), 5.46 (s, 2H), 2.40 (s, 3H) ppm. ^13^C NMR (75 MHz, CDCl_3_): *δ* = 159.2 (d, *J* = 241.2 Hz, C_quat_), 139.2 (C_quat_), 135.4 (C_quat_), 133.7 (C_quat_), 131.8 (2Ar-*C*H), 129.4 (2Ar-*C*H), 129.3 (2Ar-*C*H), 128.6 (Ar-*C*H), 128.1 (d, *J* = 10.4 Hz, C_quat_), 126.9 (2Ar-*C*H), 119.7 (C_quat_), 116.2 (d, *J* = 27.0 Hz, Ar-*C*H), 114.0 (C_quat_), 112.5 (C_quat_), 112.3 (d, *J* = 9.1 Hz, Ar-*C*H), 109.8 (d, *J* = 5.4 Hz, C_quat_), 106.5 (d, *J* = 24.0 *Hz*), 97.8 (C_quat_), 78.4 (C_quat_), 49.9 (*C*H_2_), 21.7 (*C*H_3_) ppm. ^19^F NMR (282 MHz, CDCl_3_): *δ* = −120.3 ppm. HRMS (ESI): calcd. for C_25_H_18_FN_2_ (M + H)^+^ 365.14540; found 365.14514.

#### 3.2.7. General Procedure for Suzuki Coupling (Series 9)

A Schlenk tube was charged with toluene (3 mL), ethanol (2 mL), sat. aq. NaHCO_3_ (1.5 mL), compound **7a** (300 mg, 0.83 mmol, 1 equiv.) and boronic acid (1.2 mmol, 1.5 equiv.). The tube was vigorously stirred for few minutes and Pd(PPh_3_)_4_ (90 mg, 0.083 mmol, 10 mol%) was added to the solution. The mixture was heated at 130 °C for 4 h and then cooled to room temperature. The mixture was extracted with EtOAc (3 × 10 mL), and the combined organic layers were washed with water (2 × 10 mL) and dried over MgSO_4_. Solvents were removed under reduced pressure, and the residue was purified by column chromatography using petroleum ether/EtOAc as eluent (80:20).

*1-Benzyl-3-(4-methoxyphenyl)-1H-indole-2-carbonitrile* (**9a**): Brown solid, 79%, 221 mg. mp 153–155 °C. ^1^H NMR (300 MHz, CDCl_3_): *δ* = 7.85 (dt, *J* = 8.2 Hz, *J* = 0.9 Hz, 1H, Ar-*H*), 7.67 (d, *J* = 8.9 *Hz*, 2H, Ar-*H*), 7.39–7.35 (m, 2H, Ar-*H*), 7.33–7.28 (m, 3H, Ar-*H*), 7.26–7.20 (m, 3H, Ar-*H*), 7.07 (d, *J* = 8.9 *Hz*, 2H, Ar-*H*), 5.52 (s, 2H, CH_2_), 3.89 (s, 3H, OC*H*_3_) ppm. ^13^C NMR (75 MHz, CDCl_3_): *δ* = 159.7 (C_quat_), 138.0 (C_quat_), 136.2 (C_quat_), 130.1 (2Ar-*C*H), 129.1 (2Ar-*C*H), 128.5 (C_quat_), 128.2 (2Ar-*C*H), 127.1 (2Ar-*C*H), 126.5 (Ar-*C*H), 125.3 (C_quat_), 124.4 (C_quat_), 121.8 (2Ar-*C*H), 114.7 (2Ar-*C*H), 114.5 (C_quat_), 110.9 (Ar-*C*H), 107.2 (C_quat_), 55.5 (O*C*H_3_), 49.2 (*C*H_2_) ppm. HRMS (ESI): calcd. for C_23_H_19_N_2_O (M + H)^+^ 339.14919; found 339.14965.

*1-Benzyl-3-(4-fluorophenyl)-1H-indole-2-carbonitrile* (**9b**): Brown solid, 81%, 220 mg. mp 120–122 °C. ^1^H NMR (300 MHz, CDCl_3_): *δ* = 7.82 (dt, *J* = 8.2 Hz, *J* = 0.9 Hz, 1H, Ar-*H*), 7.70 (dd, *J* = 8.8 Hz, *J* = 5.3 Hz, 2H, Ar-*H*), 7.42–7.39 (m, 2H, Ar-*H*), 7.37–7.27 (m, 3H, Ar-*H*), 7.25–7.19 (m, 3H, Ar-*H*), 5.53 (s, 2H, C*H*_2_) ppm. ^13^C NMR (75 MHz, CDCl_3_): *δ* = 162.6 (d, *J* = 248.0 Hz, C_quat_), 137.9 (C_quat_), 136.0 (C_quat_), 130.6 (d, *J* = 8.1 Hz, Ar-*C*H), 129.1 (2Ar-*C*H), 128.3 (Ar-*C*H), 127.9 (d, *J* = 3.3 Hz, C_quat_), 127.5 (C_quat_), 127.0 (2Ar-*C*H), 126.6 (Ar-*C*H), 125.1 (C_quat_), 122.1 (Ar-*C*H), 121.4 (Ar-*C*H), 116.2 (d, *J* = 21.6 Hz, Ar-*C*H), 114.1 (C_quat_), 111.0 (Ar-*C*H), 107.5 (C_quat_), 49.2 (*C*H_2_) ppm. ^19^F NMR (282 MHz, CDCl_3_): *δ* = −113.2 ppm. HRMS (ESI): calcd. for C_22_H_16_FN_2_ (M + H)^+^ 327.12975; found 327.12988.

*1-Benzyl-3-(4-(tert-butyl)phenyl)-1H-indole-2-carbonitrile* (**9c**): Brown solid, 83%, 252 mg. mp 149–151 °C. ^1^H NMR (300 MHz, CDCl_3_): *δ* = 7.92 (dt, *J* = 8.2 Hz, *J* = 0.8 Hz, 1H, Ar-*H*), 7.72 (d, *J* = 8.5 Hz, 2H, Ar-*H*), 7.57 (d, *J* = 8.5 Hz, 2H, Ar-*H*), 7.40–7.37 (m, 2H, Ar-*H*), 7.35–7.207 (m, 6H, Ar-*H*), 5.51 (s, 2H, C*H*_2_), 1.41 (s, 9H, C(C*H*_3_)_3_) ppm. ^13^C NMR (75 MHz, CDCl_3_): *δ* = 151.2 (C_quat_), 138.0 (C_quat_), 136.2 (C_quat_), 129.1 (2Ar-*C*H), 129.0 (C_quat_), 128.6 (C_quat_), 128.5 (2Ar-*C*H), 128.2 (Ar-*C*H), 127.0 (2Ar-*C*H), 126.5 (Ar-*C*H), 126.1 (2Ar-*C*H), 125.0 (C_quat_), 121.9 (Ar-*C*H), 121.8 (Ar-*C*H), 114.5 (C_quat_), 111.0 (Ar-*C*H), 107.4 (C_quat_), 49.1 (*C*H_2_), 34.8 (C_quat_), 31.4 (C(*C*H_3_)_3_) ppm. HRMS (ESI): calcd. for C_26_H_25_N_2_ (M + H)^+^ 365.20177; found 365.20152.

*1-Benzyl-3-(4-(methylthio)phenyl)-1H-indole-2-carbonitrile* (**9d**): Yellow viscous liquid, 89%, 263 mg. ^1^H NMR (300 MHz, CDCl_3_): *δ* = 7.85 (dd, *J* = 8.2 Hz, *J* = 0.9 Hz, 1H, Ar-*H*), 7.67 (d, *J =* 8.6 Hz, 2H, Ar-*H*), 7.42–7.38 (m, 4H, Ar-*H*), 7.36–7.29 (m, 3H, Ar-*H*), 7.26–7.22 (m, 3H, Ar-*H*), 5.53 (s, 2H, C*H*_2_), 2.55 (s, 3H, C*H*_3_) ppm. ^13^C NMR (75 MHz, CDCl_3_): *δ* = 138.8 (C_quat_), 138.0 (C_quat_), 136.1 (C_quat_), 129.2 (2Ar-*C*H), 129.1 (2Ar-*C*H), 128.6 (C_quat_), 128.3 (Ar-CH), 128.1 (C_quat_), 127.1 (4Ar-C*H*), 126.6 (Ar-C*H*), 126.6 (Ar-C*H*), 125.1 (C_quat_), 122.0 (Ar-C*H*), 121.7 (Ar-C*H*), 114.3 (C_quat_), 111.0 (Ar-*C*H), 107.4 (C_quat_), 49.2 (*C*H_2_), 15.8 (*C*H_3_) ppm. HRMS (ESI): calcd. for C_23_H_19_N_2_S (M + H)^+^ 355.12635; found 355.12680.

*1-Benzyl-3-(4-(trifluoromethyl)phenyl)-1H-indole-2-carbonitrile* (**9e**): Brown liquid, 87%, 272 g. ^1^H NMR (300 MHz, CDCl_3_): *δ* = 7.86 (d, *J* = 8.5 Hz, 2H, Ar-*H*), 7.85 (dd, *J* = 8.2 Hz, *J* = 0.9 Hz, 1H, Ar-*H*), 7.79 (d, *J* = 8.5 Hz, 2H, Ar-*H*), 7.45–7.42 (m, 2H, Ar-*H*), 7.38–7.28 (m, 4H, Ar-*H*), 7.27–7.22 (m, 2H, Ar-*H*), 5.55 (s, 2H, C*H*_2_) ppm. ^13^C NMR (75 MHz, CDCl_3_): *δ* = 137.9 (C_quat_), 135.8 (C_quat_), 135.7 (q, *J* = 1.2 Hz, C_quat_), 130.0 (q, *J* = 32.7 Hz, C_quat_), 129.2 (2Ar-*C*H), 129.1 (2Ar-*C*H), 128.4 (Ar-*C*H), 127.1 (2Ar-*C*H), 126.9 (Ar-*C*H), 126.7 (C_quat_), 126.1 (q, *J* = 3.8 Hz, 2Ar-*C*H), 124.9 (C_quat_), 124.2 (q, *J* = 272.2 Hz, C_quat_), 122.5 (Ar-*C*H), 121.3 (Ar-*C*H), 113.8 (C_quat_), 111.2 (Ar-*C*H), 108.1 (C_quat_), 49.3 (*C*H_2_) ppm. ^19^F NMR (282 MHz, CDCl_3_): *δ* = −62.5 ppm. HRMS (ESI): calcd. for C_23_H_16_F_3_N_2_ (M + H)^+^ 377.12656; found 377.12516.

*1-Benzyl-3-(2-methoxyphenyl)-1H-indole-2-carbonitrile* (**9f**): Brown solid, 89%, 250 mg. mp 132–134 °C. ^1^H NMR (300 MHz, CDCl_3_): *δ* = 7.91 (dd, *J* = 8.2 Hz, *J* = 0.9 Hz, 1H, Ar-*H*), 7.52–7.40 (m, 4H, Ar-*H*), 7.37–7.27 (m, 5H, Ar-*H*), 7.25–7.23 (m, 2H, Ar-*H*), 6.99 (ddd, *J* = 8.2 Hz, *J* = 2.6 Hz, *J* = 1.0 Hz, 1H, Ar-*H*), 5.54 (s, 2H, C*H*_2_), 3.90 (s, 3H, OC*H*_3_) ppm. ^13^C NMR (75 MHz, CDCl_3_): *δ* = 160.2 (C_quat_), 138.0 (C_quat_), 136.1 (C_quat_), 133.2 (C_quat_), 130.2 (Ar-*C*H), 129.1 (2Ar-*C*H), 128.4 (C_quat_), 128.3 (Ar-*C*H), 127.0 (2Ar-*C*H), 126.6 (Ar-*C*H), 125.2 (C_quat_), 122.0 (Ar-*C*H), 121.8 (Ar-*C*H), 121.3 (Ar-*C*H), 114.2 (Ar-*C*H), 114.1 (Ar-*C*H), 111.0 (Ar-*C*H), 109.1 (C_quat_), 107.7 (C_quat_), 55.5 (*C*H_2_), 49.2 (O*C*H_3_) ppm. HRMS (ESI): calcd. for C_23_H_19_N_2_O (M + H)^+^ 339.14919; found 339.14959.

*1-Benzyl-3-(2-fluorophenyl)-1H-indole-2-carbonitrile* (**9g**): White solid, 92%, 250 mg. mp 151–153 °C. ^1^H NMR (300 MHz, CDCl_3_): *δ* = 7.69 (ddt, *J* = 8.2 Hz, *J* = 2.2 Hz, *J* = 1.0 Hz, 1H, Ar-*H*), 7.62 (m, 1H, Ar-*H*), 7.48–7.39 (m, 4H, Ar-*H*), 7.37–7.29 (m, 4H, Ar-*H*), 7.28–7.21 (m, 3H, Ar-*H*), 5.55 (s, 2H, C*H*_2_) ppm. ^13^C NMR (75 MHz, CDCl_3_): *δ* = 159.9 (d, *J* = 249.1 Hz, C_quat_), 137.6 (C_quat_), 136.0 (C_quat_), 131.7 (d, *J* = 3.1 Hz, Ar-*C*H), 130.2 (d, *J* = 8.2 Hz, Ar-*C*H), 129.1 (2Ar-*C*H), 128.3 (Ar-*C*H), 127.1 (2Ar-*C*H), 126.5 (Ar-*C*H), 125.7 (C_quat_), 124.7 (d, *J* = 3.6 Hz, Ar-*C*H), 122.0 (d, *J* = 3.2 Hz, Ar-*C*H), 122.0 (Ar-*C*H), 119.6 (d, *J* = 15.4 Hz, C_quat_), 116.5 (d, *J* = 22.1 Hz, Ar-*C*H), 113.6 (C_quat_), 111.9 (Ar-*C*H), 109.1 (C_quat_), 104.5 (C_quat_), 49.4 (*C*H_2_) ppm. ^19^F NMR (282 MHz, CDCl_3_): *δ* = −112.1 ppm. HRMS (ESI): calcd. for C_22_H_16_FN_2_ (M + H)^+^ 327.12975; found 327.13012.

*1-Benzyl-3-(o-tolyl)-1H-indole-2-carbonitrile* (**9h**): Brown solid, 92%, 246 mg. mp 130–132 °C. ^1^H NMR (300 MHz, CDCl_3_): *δ* = 7.49 (dt, *J* = 8.1 Hz, *J* = 1.0 Hz, 1H, Ar-*H*), 7.42–7.37 (m, 5H, Ar-*H*), 7.36–7.32 (m, 4H, Ar-*H*), 7.27–7.18 (m, 3H, Ar-*H*), 5.55 (s, 2H, *CH_2_*), 2.31 (s, 3H, C*H*_3_) ppm. ^13^C NMR (75 MHz, CDCl_3_): *δ* = 137.5 (C_quat_), 137.3 (C_quat_), 136.2 (C_quat_), 131.0 (Ar-*C*H), 130.9 (C_quat_), 130.7 (Ar-*C*H), 129.1 (2Ar-*C*H), 128.7 (C_quat_), 128.6 (Ar-*C*H), 128.2 (Ar-*C*H), 127.0 (2Ar-CH), 126.4 (Ar-*C*H), 126.3 (C_quat_), 126.0 (Ar-*C*H), 122.0 (Ar-*C*H), 121.6 (Ar-*C*H), 113.8 (C_quat_), 110.9 (Ar-*C*H), 108.9 (C_quat_), 49.2 (*C*H_2_), 20.3 (*C*H_3_) ppm. HRMS (ESI): calcd. for C_23_H_19_N_2_ (M + H)^+^ 323.15428; found 323.15463.

*1-Benzyl-3-(3-methoxyphenyl)-1H-indole-2-carbonitrile* (**9i**): Solid orange, 86%, 241 mg. mp 131–133 °C. ^1^H NMR (300 MHz, CDCl_3_): *δ* = 7.91 (d, *J* = 8.2 Hz, 1H, Ar-*H*), 7.46 (t, *J* = 7.9 Hz, 1H, Ar-*H*), 7.41–7.39 (m, 2H, Ar-*H*), 7.37–7.27 (m, 5H, Ar-*H*), 7.25–7.23 (m, 2H, Ar-*H*), 6.99 (ddd, *J* = 8.2 Hz, *J* = 2.5 Hz, *J* = 0.8 Hz*,* 1H, Ar-*H*), 5.54 (s, 2H, C*H*_2_), 3.90 (s, 3H, OC*H*_3_) ppm. ^13^C NMR (75 MHz, CDCl_3_): *δ* = 160.2 (C_quat_), 138.0 (C_quat_), 136.1 (C_quat_), 133.2 (C_quat_), 130.2 (Ar-*C*H), 129.1 (2Ar-*C*H), 128.4 (C_quat_), 128.3 (Ar-*C*H), 127.1 (2Ar-*C*H), 126.6 (Ar-*C*H), 125.2 (C_quat_), 122.0 (Ar-*C*H), 121.8 (Ar-*C*H), 121.3 (Ar-*C*H), 114.2 (C_quat_), 114.2 (Ar-*C*H), 114.1 (Ar-*C*H), 111.0 (Ar-*C*H), 107.7 (C_quat_), 55.5 (*C*H_3_), 49.2 (*C*H_2_) ppm. HRMS (ESI): calcd. for C_23_H_19_N_2_O (M + H)^+^ 339.14919; found 339.14965.

*1-Benzyl-3-(2,4-dimethoxyphenyl)-1H-indole-2-carbonitrile* (**9j**): Brown liquid, 89%, 248 mg. ^1^H NMR (300 MHz, CDCl_3_): *δ* = 7.64 (dt, *J* = 8.1 Hz, *J* = 0.9 Hz*,* 2H, Ar-*H*), 7.42 (d, *J* = 8.8 Hz, 1H, Ar-*H*), 7.34–7.27 (m, 5H, Ar-*H*), 7.24–7.14 (m, 3H, Ar-*H*), 6.63 (s, 1H, Ar-H), 6.62 (dd, *J* = 7.6 Hz, *J* = 2.4 Hz, 1H, Ar-H), 5.49 (s, 2H, C*H*_2_), 3.87 (s, 3H, OC*H*_3_), 3.84 (s, 3H, OC*H*_3_) ppm. ^13^C NMR (75 MHz, CDCl_3_): *δ* = 161.3 (C_quat_), 158.1 (C_quat_), 137.6 (C_quat_), 136.4 (C_quat_), 132.2 (Ar-*C*H), 129.0 (2Ar-*C*H), 128.1 (Ar-*C*H), 127.1 (2Ar-*C*H), 126.2 (C_quat_), 126.0 (Ar-*C*H), 124.7 (C_quat_), 122.4 (Ar-*C*H), 121.3 (Ar-*C*H), 114.3 (C_quat_), 113.3 (C_quat_), 110.7 (Ar-*C*H), 109.1 (C_quat_), 104.9 (Ar-*C*H), 99.2 (Ar-*C*H), 55.6 (O*C*H_3_), 55.4 (O*C*H_3_), 49.2 (*C*H_2_) ppm. HRMS (ESI): calcd. for C_24_H_21_N_2_O_2_ (M + H)^+^ 369.16030; found 369.16135.

*1-Benzyl-3-(2,3-dihydrobenzo[b]*[1,4]*dioxin-6-yl)-1H-indole-2-carbonitrile* (**9k**): Brown solid, 90%, 273 mg. mp 140–142 °C. ^1^H NMR (300 MHz, CDCl_3_): *δ* = 7.87 (d, *J* = 8.2 Hz, 2H, Ar-*H*), 7.38–7.36 (m, 2H, Ar-*H*), 7.35–7.27 (m, 4H, Ar-*H*), 7.25–7.21 (m, 4H, Ar-*H*), 7.02 (d, *J* = 8.3 Hz, 2H, Ar-*H*), 5.51 (s, 2H, C*H*_2_), 4.32 (s, 4H, 2 x C*H*_2_) ppm. ^13^C NMR (75 MHz, CDCl_3_): *δ* = 144.0 (C_quat_), 143.7 (C_quat_), 137.8 (C_quat_), 136.1 (C_quat_), 129.0 (2Ar-*C*H), 128.1 (Ar-*C*H), 128.1 (C_quat_), 127.0 (2Ar-*C*H), 126.4 (Ar-*C*H), 125.1 (2C_quat_), 122.0 (Ar-*C*H), 121.8 (Ar-*C*H), 121.7 (Ar-*C*H), 118.0 (Ar-*C*H), 117.6 (Ar-*C*H), 114.3 (C_quat_), 110.8 (Ar-*C*H), 107.2 (C_quat_), 64.5 (*C*H_2_), 64.4 (*C*H_2_), 49.0 (*C*H_2_) ppm. HRMS (ESI): calcd. for C_24_H_19_N_2_O_2_ (M + H)^+^ 367.14465; found 367.14425.

*1-Benzyl-3-(6-methoxynaphthalen-2-yl)-1H-indole-2-carbonitrile* (**9l**): Brown solid, 84%, 272 mg. mp 147–149 °C. ^1^H NMR (300 MHz, CDCl_3_): *δ* = 8.13 (s, 1H, Ar-*H*), 7.96–7.81 (m, 4H, Ar-*H*), 7.41–7.19 (m, 10H, Ar-*H*), 5.52 (s, 2H, C*H*_2_), 3.94 (s, 3H, OC*H*_3_) ppm. ^13^C NMR (75 MHz, CDCl_3_): *δ* = 158.3 (C_quat_), 138.0 (C_quat_), 136.2 (C_quat_), 134.3 (C_quat_), 129.8 (Ar-*C*H), 129.2 (C_quat_), 129.1 (2Ar-*C*H), 128.7 (C_quat_), 128.2 (Ar-*C*H), 127.8 (Ar-*C*H), 127.7 (Ar-*C*H), 127.2 (Ar-*C*H), 127.1 (C_quat_), 127.0 (2Ar-*C*H), 126.6 (Ar-*C*H), 125.4 (C_quat_), 121.9 (Ar-*C*H), 121.8 (Ar-*C*H), 119.5 (Ar-*C*H), 114.4 (C_quat_), 111.0 (Ar-*C*H), 107.6 (C_quat_), 105.8 (Ar-*C*H), 55.5 (O*C*H_3_), 49.2 (*C*H_2_) ppm. HRMS (ESI): calcd. for C_27_H_21_N_2_O (M + H)^+^ 389.16484; found 389.16527.

*1-Benzyl-3-(2-methoxypyridin-3-yl)-1H-indole-2-carbonitrile* (**9m**): Brown liquid, 86%, 241 mg. ^1^H NMR (300 MHz, CDCl_3_): *δ* = 8.27 (dd, *J* = 5.0 Hz, *J* = 1.9 Hz, 1H, Ar-*H*), 7.83 (dd, *J* = 7.3 Hz, *J* = 1.9 Hz, 1H, Ar-*H*), 7.63 (d, *J* = 8.2 Hz, 2H, Ar-*H*), 7.39–7.37 (m, 3H, Ar-*H*), 7.34–7.29 (m, 3H, Ar-*H*), 7.27–7.19 (m, 2H, Ar-*H*), 7.05 (dd, *J* = 7.3 Hz, *J* = 5.1 Hz, 1H, Ar-*H*), 5.54 (s, 2H, C*H*_2_), 4.03 (s, 3H, OC*H*_3_) ppm. ^13^C NMR (75 MHz, CDCl_3_): *δ* = 161.3 (C_quat_), 147.0 (Ar-*C*H), 139.9 (Ar-*C*H), 137.6 (C_quat_), 136.1 (C_quat_), 129.1 (2Ar-*C*H), 128.3 (Ar-*C*H), 127.1 (2Ar-*C*H), 126.4 (Ar-*C*H), 125.8 (C_quat_), 122.8 (C_quat_), 122.1 (Ar-*C*H), 121.8 (Ar-*C*H), 117.0 (Ar-*C*H), 115.3 (C_quat_), 113.7 (C_quat_), 111.0 (Ar-*C*H), 109.4 (C_quat_), 53.6 (*C*H_2_), 49.4 (O*C*H_3_) ppm. HRMS (ESI): calcd. for C_22_H_18_N_3_O (M + H)^+^ 340.14499; found 340.14471.

*1-Benzyl-5-methoxy-3-(4-methoxyphenyl)-1H-indole-2-carbonitrile* (**9n**): Brown solid, 89%, 254 mg. mp 152–154 °C. ^1^H NMR (300 MHz, CDCl_3_): *δ* = 7.65 (d, *J* = 8.8 Hz, 2H, Ar-*H*), 7.37–7.27 (m, 3H, Ar-*H*), 7.26–7.18 (m, 4H, Ar-*H*), 7.10–7.02 (m, 3H, Ar-*H*), 5.48 (s, 2H, C*H*_2_), 3.89 (s, 3H, OC*H*_3_), 3.82 (s, 3H, OC*H*_3_) ppm. ^13^C NMR (75 MHz, CDCl_3_): *δ* = 159.5 (C_quat_), 155.7 (C_quat_), 136.3 (C_quat_), 133.3 (C_quat_), 129.9 (2Ar-*C*H), 129.1 (2Ar-*C*H), 128.2 (Ar-*C*H), 127.6 (C_quat_), 127.0 (2Ar-*C*H), 125.6 (C_quat_), 124.5 (C_quat_), 118.0 (Ar-*C*H), 114.7 (2Ar-*C*H), 114.5 (C_quat_), 111.9 (Ar-*C*H), 107.5 (C_quat_), 101.6 (Ar-*C*H), 55.9 (O*C*H_3_), 55.5 (O*C*H_3_), 49.3 (*C*H_2_) ppm. HRMS (ESI): calcd. for C_24_H_21_N_2_O_2_ (M + H)^+^ 369.15975; found 369.16033.

*1-Benzyl-6-methoxy-3-(3-methoxyphenyl)-1H-indole-2-carbonitrile* (**9o**): Brown solid, 93%, 264 mg. mp 150–152 °C. ^1^H NMR (300 MHz, CDCl_3_): *δ* = 7.78 (d, *J* = 8.9 Hz, 1H, Ar-*H*), 7.46 (t, *J* = 7.9 Hz, 1H, Ar-*H*), 7.41–7.22 (m, 7H, Ar-*H*), 6.99 (dd, *J* = 7.9 Hz, *J* = 2.1 Hz, 1H, Ar-*H*), 6.93 (dd, *J* = 8.9 Hz, *J* = 2.1 Hz, 1H, Ar-*H*), 6.75 (d, *J* = 2.1 Hz, 1H, Ar-*H*), 5.49 (s, 2H, C*H*_2_), 3.91 (s, 3H, OC*H*_3_), 3.84 (s, 3H, OC*H*_3_) ppm. ^13^C NMR (75 MHz, CDCl_3_): *δ* = 160.2 (C_quat_), 159.8 (C_quat_), 139.1 (C_quat_), 136.1 (C_quat_), 133.3 (C_quat_), 130.2 (Ar-*C*H), 129.1 (2Ar-*C*H), 128.8 (C_quat_), 128.2 (Ar-*C*H), 127.0 (2Ar-*C*H), 122.6 (Ar-*C*H), 121.2 (Ar-*C*H), 119.5 (C_quat_), 114.6 (C_quat_), 114.2 (Ar-*C*H), 114.0 (Ar-*C*H), 113.1 (Ar-*C*H), 106.4 (C_quat_), 93.1 (Ar-*C*H), 55.7 (O*C*H_3_), 55.5 (O*C*H_3_), 49.1 (*C*H_2_) ppm. HRMS (ESI): calcd. for C_24_H_21_N_2_O_2_ (M + H)^+^ 369.15975; found 369.16017.

*1-Benzyl-5-fluoro-3-(3-methoxyphenyl)-1H-indole-2-carbonitrile* (**9p**): Yellow viscous liquid, 98%, 279 mg. ^1^H NMR (300 MHz, CDCl_3_): *δ* = 7.51 (dd, *J =* 7.4 *Hz*, *J* = 1.6 *Hz*, 1H), 7.44 (td, *J* = 8.1 *Hz*, *J* = 1.7 Hz, 1H), 7.36–7.23 (m, 1H), 7.16–7.08 (m, 3H), 5.52 (s, 2H), 3.90 (s, 3H) ppm. ^13^C NMR (75 MHz, CDCl_3_): *δ* = 158.7 (d, *J* = 238.7 Hz, C_quat_), 156.9 (C_quat_), 136.0 (C_quat_), 134.1 (C_quat_), 131.4 (Ar-*C*H), 130.0 (Ar-*C*H), 129.1 (2Ar-*C*H), 128.3 (Ar-*C*H), 127.0 (2Ar-*C*H), 126.4 (d, *J =* 10.5 Hz, C_quat_), 124.2 (d, *J =* 5.5 Hz, C_quat_), 121.0 (Ar-*C*H), 120.1 (C_quat_), 115.2 (d, *J* = 26.8 Hz, Ar-*C*H), 113.7 (C_quat_), 111.8, (d, *J* = 9.4 Hz, Ar-*C*H), 111.5 (Ar-*C*H), 110.8 (C_quat_), 106.8 (d, *J =* 24.1 Hz, Ar-*C*H), 55.4 (O*C*H_3_), 49.5 (*C*H_2_) ppm. ^19^F NMR (282 MHz, CDCl_3_): *δ* = −121.7 ppm. HRMS (ESI): calcd. for C_23_H_18_FN_2_O (M + H)^+^ 357.14032; found 357.14081.

#### 3.2.8. General Procedure for Heck Coupling (Series 10)

Compound **7a** (300 mg, 0.083 mmol, 1 equiv.), alkene (1.2 mmol, 1.5 equiv), KOAc (50 mg, 0.5 mmol, 6 equiv.), Pd(OAc)_2_ (0.74 mg, 0.003 mmol, 0.04 equiv.), tBuNH_4_Cl(46 mg, 0.166 mmol, 2 equiv.) and DMF (5 mL) were introduced in a sealed tube. The tube was vacuumed and backfilled with argon. The reaction mixture was stirred at 80 °C for 24 h and then cooled to room temperature. The reaction mixture was quenched with water, filtered through a Celite^©^ pad and washed with EtOAc (100 mL). Then, the resulting organic layer was washed with sat. aq. NH_4_Cl (2 × 40 mL) and brine (3 × 40 mL). The organic layers were dried over MgSO_4_ and concentrated under reduced pressure. The crude product was purified by column chromatography using petroleum ether/EtOAc as eluent (70:30).

*(E)-Methyl-3-(1-benzyl-2-cyano-1H-indol-3-yl) acrylate* (**10a**): Brown solid, 92%, 244 mg. ^1^H NMR (300 MHz, CDCl_3_): *δ* = 7.99 (d, *J* = 16.2 Hz, 1H, C*H*=CH), 7.95 (dt, *J* = 8.2 Hz, *J* = 0.9 Hz, 1H, Ar-*H*), 7.46–7.30 (m, 6H, Ar-*H*), 7.18 (m, 2H, Ar-*H*), 6.76 (d, *J* = 16.2 Hz, 1H, CH=C*H*), 5.50 (s, 2H, C*H*_2_), 3.85 (s, 3H, OC*H*_3_) ppm. ^13^C NMR (75 MHz, CDCl_3_): *δ* = 167.5 (C_quat_), 138.1 (C_quat_), 135.3 (C_quat_), 134.0 (*C*H=CH), 129.2 (2Ar-*C*H), 128.5 (Ar-*C*H), 127.1 (Ar-*C*H), 127.0 (2Ar-C*H*), 124.9 (C_quat_), 123.2 (Ar-*C*H), 121.8 (C_quat_), 121.6 (Ar-*C*H), 119.1 (CH=*C*H), 112.8 (C_quat_), 111.7 (C_quat_), 111.5 (Ar-*C*H), 52.0 (O*C*H_3_), 49.6 (*C*H_2_) ppm. HRMS (ESI): calcd. for C_20_H_17_N_2_O_2_ (M + H)^+^ 317.12900; found 317.12946.

(*E*)-1-Benzyl-3-styryl-1*H*-indole-2-carbonitrile (**10b**): Brown solid, 89%, 257 mg. ^1^H NMR (300 MHz, CDCl_3_): *δ* = 8.06 (dt, *J* = 8.1 Hz, *J* = 0.7 Hz, 1H), 7.66–7.58 (m, 2H, Ar-*H*), 7.51 (d, *J* = 16.5 Hz, C*H*=CH), 7.45–7.29 (m, 10H, 9Ar-*H* + CH=C*H*), 7.23–7.20 (m, 2H, Ar-*H*), 5.45 (s, 2H, C*H*_2_) ppm. ^13^C NMR (75 MHz, CDCl_3_): *δ* = 138.1 (C_quat_), 137.3 (C_quat_), 135.9 (C_quat_), 131.2 (*C*H=CH), 129.1 (2Ar-*C*H), 128.9 (2Ar*C*H), 128.2 (Ar-*C*H), 128.1 (Ar-*C*H), 126.9 (2Ar-*C*H), 126.6 (Ar-*C*H), 126.5 (2Ar-*C*H), 125.2 (C_quat_), 124.6 (C_quat_), 122.0 (CH=*C*H), 121.7 (Ar-*C*H), 118.4 (Ar-*C*H), 113.9 (C_quat_), 111.0 (Ar-*C*H), 108.6 (C_quat_), 49.1 (*C*H_2_) ppm. HRMS (ESI): calcd. for C_24_H_19_N_2_ (M + H)^+^ 335.15482; found 335.15532.

*1-Benzyl-3-(3-cyanoprop-1-en-1-yl)-1H-indole-2-carbonitrile* (**10c**): Brown solid, 81%, 192 mg. ^1^H NMR (300 MHz, CDCl_3_): *δ* = 8.09 (d, *J* = 8.1 *Hz*, 1H, *Z*-isomer), 7.79 (d, *J* = 8.2 *Hz*, 1H, *E*-isomer), 7.61 (d, *J* = 16.7 *Hz*, 1H, *E*-isomer), 7.52–7.29 (m, Ar-CH), 7.23–7.18 (m, Ar-CH), 6.14 (d, *J* = 16.7 *Hz*, 1H, *E*-isomer), 5.64 (d, *J* = 12.1 *Hz*, 1H, *Z*-isomer), 5.48 (s, 2H) ppm. ^13^C NMR *E*-isomer (75 MHz, CDCl_3_): *δ* = 139.5 (*C*H=CH), 137.9 (C_quat_), 134.9 (C_quat_), 129.2 (2Ar-*C*H), 128.6 (Ar-*C*H), 127.4 (Ar-*C*H), 126.9 (2Ar-*C*H), 124.1 (C_quat_), 123.7 (Ar-*C*H), 120.8 (Ar-*C*H), 120.7 (C_quat_), 118.3 (C_quat_), 111.2 (C_quat_), 111.7 (Ar-*C*H), 111.3 (C_quat_), 96.9 (CH=*C*H), 49.6 (*C*H_2_) ppm. HRMS (ESI): calcd. for C_19_H_14_N_3_ (M + H)^+^ 284.11877; found 284.11842.

*(E)-Methyl-3-(1-benzyl-2-cyano-5-methoxy-1H-indol-3-yl) acrylate* (**10d**): Brown solid, 88%, 236 mg. ^1^H NMR (300 MHz, CDCl_3_): *δ* = 7.97 (d, *J* = 16.2 Hz, 1H, C*H*=CH), 7.36–7.28 (m, 3H, Ar-C*H*), 7.27–7.23 (m, 2H, Ar-C*H*), 7.16 (dd, *J* = 7.5 Hz, *J* = 2.0 Hz, 2H, Ar-C*H*), 7.06 (dd, *J* = 9.2 Hz, *J* = 2.3 Hz, 1H, Ar-C*H*), 6.69 (d, *J* = 16.2 Hz, 1H, CH=C*H*) 5.46 (s, 2H, C*H*_2_), 3.88 (s, 3H, OC*H*_3_), 3.85 (s, 3H, OC*H*_3_) ppm. ^13^C NMR (75 MHz, CDCl_3_): *δ* = 167.6 (C_quat_), 156.6 (C_quat_), 135.3 (C_quat_), 134.1 (*C*H=CH), 133.2 (C_quat_), 129.2 (2Ar-*C*H), 128.5 (Ar-*C*H), 126.9 (2Ar-*C*H), 125.5 (C_quat_), 120.8 (C_quat_), 118.4 (Ar-*C*H), 118.1 (Ar-*C*H), 112.9 (C_quat_), 112.4 (Ar-*C*H), 111.3 (C_quat_), 101.6 (Ar-*C*H), 55.9 (O*C*H_3_), 51.9 (O*C*H_3_), 49.7 (*C*H_2_) ppm. HRMS (ESI): calcd. for C_21_H_19_N_2_O_3_ (M + H)^+^ 347.13957; found 347.14053.

*(E)-Methyl-3-(1-benzyl-2-cyano-6-methoxy-1H-indol-3-yl) acrylate* (**10e**): Brown solid, 92%, 247 mg. ^1^H NMR (300 MHz, CDCl_3_): *δ* = 7.90 (d, *J* = 16.1 Hz, 1H, C*H*=CH), 7.76 (d, *J* = 9.0 Hz, 1H, Ar-C*H*), 7.35–7.28 (m, 3H, Ar-C*H*), 7.21–7.13 (m, 2H, Ar-C*H*),6.94 (dd, *J* = 8.9 Hz, *J* = 2.0 Hz, 1H, Ar-C*H*), 6.71 (s, 1H, Ar-C*H*), 6.68 (d, *J* = 16.1 Hz, 1H, CH=C*H*), 5.40 (s, 2H, C*H*_2_), 3.83 (s, 3H, OC*H*_3_), 3.80 (s, 3H, OC*H*_3_) ppm. ^13^C NMR (75 MHz, CDCl_3_): *δ* = 167.4 (C_quat_), 159.9 (C_quat_), 139.3 (C_quat_), 135.2 (C_quat_), 134.6 (C_quat_), 133.9 (*C*H=CH), 129.2 (2Ar-*C*H), 128.4 (Ar-*C*H), 126.9 (2Ar-*C*H), 122.2 (Ar-*C*H), 122.0 (C_quat_), 118.8 (Ar-*C*H), 113.9 (Ar-*C*H), 113.0 (C_quat_), 110.4 (C_quat_), 93.6 (CH=*C*H), 55.6 (O*C*H_3_), 51.8 (O*C*H_3_), 49.3 (*C*H_2_) ppm. HRMS (ESI): calcd. for C_21_H_19_N_2_O_3_ (M + H)^+^ 347.13957; found 347.14046.

*(E)-Methyl-3-(1-benzyl-2-cyano-5-fluoro-1H-indol-3-yl)acrylate* (**10f**): Brown solid, 81%, 215 mg. ^1^H NMR (300 MHz, CDCl_3_): *δ* = 7.92 (d, *J* = 16.2 Hz*,* 1H, C*H*=CH), 7.57 (dd, *J* = 9.1 *Hz*, *J* = 2.4 *Hz*, 1H, Ar-C*H*), 7.40–7.27 (m, 4H, Ar-C*H*), 7.22–7.12 (m, 3H, Ar-C*H*), 6.67 (d, *J* = 16.2 *Hz*, 1H, CH=C*H*), 5.48 (s, 2H, C*H*_2_), 3.85 (s, 3H, OC*H*_3_) ppm. ^13^C NMR (75 MHz, CDCl_3_): *δ* = 167.3 (C_quat_), 159.6 (d, *J* = 241.4 Hz, C_quat_), 134.6 (C_quat_), 133.5 (*C*H=CH), 129.3 (2Ar-CH), 128.7 (Ar-*C*H), 126.9 (2Ar-CH), 125.2 (d, *J* = 10.1 Hz, C_quat_), 121.4 (d, *J* = 5.3 Hz, C_quat_), 119.2 (CH=*C*H), 116.2 (d, *J* = 26.8 Hz, Ar-*C*H), 112.9 (C_quat_), 112.7 (d, J = 9.6 Hz, Ar-*C*H), 112.4 (C_quat_), 106.5 (d, *J* = 24.5 Hz, Ar-*C*H), 52.0 (O*C*H_3_), 49.9 (*C*H_2_) ppm. ^19^F NMR (282 MHz, CDCl_3_): *δ* = −118.71 ppm. HRMS (ESI): calcd. for C_20_H_16_FN_2_O_2_ (M + H)^+^ 335.11958; found 335.12021.

#### 3.2.9. General Procedure for Stille Coupling (Series 11)

A Schlenk flask was charged with compound **7a** (300 mg, 0.84 mmol, 1 equiv.), DMF (5 mL), vinyl tin derivatives (1.008 mmol, 1.2 equiv.) and PdCl_2_(MeCN)_2_ (21.8 mg, 0.084 mmol, 10 mol%) under argon. The mixture was stirred at 40 °C for 1 week. Then, the reaction mixture was added to a mixture of KF (2 M) and ethyl acetate (50:50), stirred for 30 min at room temperature and filtered through a Celite^©^ pad. Solvents were removed under reduced pressure, and the crude product was purified by column chromatography with a stationary phase composed of 10% powdered anhydrous K_2_CO_3_ and silica, using petroleum ether/ethyl acetate as eluent (99.9:0.1).

*(E)-1-Benzyl-3-(3-methylbut-1-en-1-yl)-1H-indole-2-carbonitrile* (**11b**): Brown oil, 35%, 88 mg. ^1^H NMR (300 MHz, CDCl_3_): *δ* = 7.90 (dt, *J* = 8.2 Hz, *J* = 0.9 Hz, Hz*,* 1*H*), 7.39–7.26 (m, 5H, Ar-C*H*), 7.25–7.13 (m, 3H, Ar-C*H*), 6.63 (d, *J* = 16.2 *Hz*, 1H, C*H*=CH), 6.54 (dd, *J* = 16.2 Hz, *J* = 6.2 Hz, 1H, CH=C*H*), 5.44 (s, 2H, C*H*_2_), 2.56 (oct, *J* = 6.6 Hz, 1H, C*H*(CH_3_)_2_), 1.16 (d, *J* = 6.7 Hz, 6H, 2xC*H*_3_) ppm. ^13^C NMR (75 MHz, CDCl_3_): *δ* = 141.8 (*C*H=CH), 138.1 (C_quat_), 136.2 (C_quat_), 129.1 (2Ar-*C*H), 128.2 (Ar-*C*H), 127.0 (C_quat_), 126.9 (2Ar-*C*H), 126.5 (Ar-*C*H), 125.8 (C_quat_), 124.8 (C_quat_), 121.8 (Ar-*C*H), 121.6 (CH=*C*H), 116.9 (Ar-*C*H), 114.1 (C_quat_), 110.9 (Ar-*C*H), 49.0 (*C*H_2_), 32.5 (*C*H(CH_3_)_2_), 22.6 (2x*C*H_3_) ppm. HRMS (ESI): calcd. for C_21_H_21_N_2_ (M + H)^+^ 301.17047; found 301.17038.

(*E*)-1-Benzyl-3-(2-(trimethylsilyl)vinyl)-1*H*-indole-2-carbonitrile (**11c**): Orange oil, 40%, 111 mg. ^1^H NMR (300 MHz, CDCl_3_): *δ* = 8.01 (dt, *J* = 8.2 Hz*,J* = 0.8 Hz, 1H, Ar-*C*H), 7.41–7.23 (m, 6H, Ar-C*H*), 7.17 (d, *J* = 19.6 Hz, 1H, C*H*=CH), 7.16 (dd, *J* = 7.9 *Hz*, *J* = 2.2 Hz, 2H, Ar-C*H*), 6.82 (d, *J* = 19.5 Hz, 1H, CH=C*H*), 5.45 (s, 2H), 0.22 (s, 9H, 3xC*H*_3_) ppm. ^13^C NMR (75 MHz, CDCl_3_): *δ* = 138.0 (C_quat_), 135.9 (C_quat_), 133.6 (*C*H=CH), 133.0 (Ar-*C*H), 129.0 (2Ar-*C*H), 128.2 (Ar-*C*H), 126.8 (2Ar-*C*H), 126.5 (Ar-*C*H), 124.7 (C_quat_), 117.0 (C_quat_), 113.8 (C_quat_), 110.9 (Ar-*C*H), 108.5 (C_quat_), 49.0 (*C*H_2_), −1.16 (3x*C*H_3_) ppm. HRMS (ESI): calcd. for C_21_H_23_N_2_Si (M + H)^+^ 331.16305; found 335.16298.

#### 3.2.10. Typical Procedure for Addition of *n*-BuLi on Compound **8b**

A suspension of compound **8b** (300 mg, 0.865 mmol) in THF (15 mL) was introduced into a dry, single-necked, round-bottomed flask under argon. The suspension was vigorously stirred, and *n*-BuLi (2 mmol, 2.2 equiv., 0.17 mL, 1.6 M in hexane) was added at −78 °C. The mixture was stirred magnetically under argon for 4h at room temperature. Then, ZnI_2_ (276mg, 0.865 mmol, 1 equiv.) and I_2_ (241 mg, 0.95 mmol, 1.1 equiv.) were added at −78 °C, and the suspension was stirred overnight under argon at rt. The mixture was basified with 2 N aq NaOH (10 mL), extracted with EtOAc (10 mL) and washed with 5% aq Na_2_S_2_O_3_ (10 mL). The organic layer was dried (MgSO_4_), filtered and concentrated under reduced pressure. The residue was purified by chromatography on silica gel.

*1-(1-Benzyl-3-(p-tolylethynyl)-1H-indol-2-yl)pentan-1-one* (**8b’**): Orange solid, 78.0%, 281 mg. ^1^H NMR (300 MHz, CDCl_3_): *δ* = 7.95 (dt, *J* = 7.9 Hz*,J* = 0.9 Hz, 1H, Ar-*C*H), 7.51–7.40 (m, 2H, Ar-*C*H), 7.30–7.20 (m, 6H, Ar-*C*H), 7.05–7.02 (m, 2H, Ar-*C*H), 5.84 (s, 2H, CH_2_), 3.39 (t, *J* = 7.3 Hz*,* 2H, CH_2_), 2.41 (s, 3H, CH_3_), 1.72 (q, *J* = 7.5 Hz, 2H, CH_2_), 1.37 (sext, *J* = 7.6 Hz, 2H, CH_2_), 0.91 (t, *J* = 7.3 Hz*,* 3H, CH_3_) ppm. ^13^C NMR (75 MHz, CDCl_3_): *δ* = 195.2 (C_quat_), 138.8 (C_quat_), 138.7 (C_quat_), 138.2 (C_quat_), 135.8 (C_quat_), 131.3 (2Ar-*C*H), 129.4 (2Ar-*C*H), 128.7 (2Ar-*C*H), 128.5 (C_quat_), 127.3 (Ar-*C*H), 126.9 (Ar-*C*H), 126.5 (2Ar-*C*H), 122.2 (Ar-*C*H), 121.8 (Ar-*C*H), 120.5 (C_quat_), 111.1 (Ar-*C*H), 105.9 (C_quat_), 98.5 (C_quat_), 85.8 (C_quat_), 48.7 (*C*H_2_), 42.4 (*C*H_2_), 27.0 (*C*H_2_), 22.6 (*C*H_2_), 21.7 (*C*H_3_), 14.1 (*C*H_3_) ppm. HRMS (ESI): calcd. for C_29_H_28_NO (M + H)^+^ 406.21709; found 406.21720.

## 4. Conclusions

In summary, we have developed a palladium-catalyzed homocoupling reaction of heteroatoms through C–I bond functionalization using different alkynes, alkenes and aryl derivatives. This approach offers a simple strategy and alternative route for the preparation of heteroaryl nitriles from easily available precursors in good to excellent yields. The present work allows access to novel diversely polysubstituted cyanoindoles via Sonogashira, Suzuki–Miyaura, Heck and Stille cross-coupling reactions.

## Data Availability

The data presented in this sudy ara available in Appendix A.

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
