# Peer review of "Synthesis of New Highly Functionalized 1H-Indole-2-carbonitriles via Cross-Coupling Reactions"

_molecules, 2021, doi:10.3390/molecules26175287_

Round 1

Reviewer 1 Report

In this contribution, Hrizi et al. present a work on the synthesis of indole compounds that can serve as building blocks for synthesis of drug-like species. In particular, a series of 1-(but-2-ynyl)-1H-indole-2-carbonitrile and 1-benzyl-3-iodo-1H-indole-2-carbonitrile compounds were prepared and its use of precursor on cross-coupling Sonogahsira, Susuki and Heck reactions is analyzed.

The work is strong on synthetic aspects and compound characterization, and will well received for the readers of the Special Issue of Molecule called "Preparation of Heterocycles by Metal-Promoted Reactions".

Thus, I recommend publication in Molecules after minor revision.

The broad interest of cross-coupling methodology can be highlighted by including a few examples on the reaction scope. Plenty of examples including applications can be given: Journal Molecular Structure, 2017, 1139, 353, Journal Organometallic Chem. 634, 2001, 39.

Author Response

First of all, we would like to express our gratitude to the referees for their careful examination of our work. We have considered the reviewer’s comments on our manuscript carefully and have made some changes to address the comments of reviewer’s where appropriate. We will now give a detailed response below:

Reviewer 1

In this contribution, Hrizi et al. present a work on the synthesis of indole compounds that can serve as building blocks for synthesis of drug-like species. In particular, a series of 1-(but-2-ynyl)-1H-indole-2-carbonitrile and 1-benzyl-3-iodo-1H-indole-2-carbonitrile compounds were prepared and its use of precursor on cross-coupling Sonogahsira, Susuki and Heck reactions is analyzed.

The work is strong on synthetic aspects and compound characterization, and will well received for the readers of the Special Issue of Molecule called "Preparation of Heterocycles by Metal-Promoted Reactions".

Thus, I recommend publication in Molecules after minor revision.

The broad interest of cross-coupling methodology can be highlighted by including a few examples on the reaction scope. Plenty of examples including applications can be given: Journal Molecular Structure, 2017, 1139, 353, Journal Organometallic Chem. 634, 2001, 39.

Thank you very much for your careful examination of our work and suggestions. Based on your comments, we deeply improved our revised manuscript. The answers to your comments and changes made are listed below.

The following sentence has been added in this revision :

The broad interest of this cross-coupling methodology is thus found in many fields of application [42,43].

Consequently, the reference 42 and 43 were both added in this revision :

  1. Khairul, W.M.; Daud, A.I.; Hanifaah, N.A.M.; Arshad, S.; Razak, I.A.; Zuki, H. M.; Erben, M.F. Structural study of a novel acetylide-thiourea derivative and its evaluation as a detector of benzene. J. Mol. Struct. 2017, 1139, 353–361.
  2. Buchmeiser, M.R.; Schareina, T.; Kempe, R.; Wurst, K. Bis(pyrimidine)-based palladium catalysts: synthesis, X-ray structure and applications in Heck–, Suzuki–, Sonogashira–Hagihara couplings and amination reactions. J. Organomet. Chem. 2001, 634, 39–46.

Please note that all corrections and changes made have been highlighted in yellow in this revision.

Reviewer 2 Report

The manuscript “Synthesis of New and Highly Functionalized 1H-indole-2-carbonitriles via Cross-Coupling Reactions” describes the syntheses of indole derivatives using cross-coupling reactions.

Biological assays were not performed for the compounds synthesized. The manuscript deserves publication after revision. Listed below are some suggestions for the authors to consider.

General comment: It is not indicated where introduction ends, and results and discussions start. After that “Conclusion” is shown. Clear separation of the sections may be indicated.

Abstract: A study of the reactivity of iodine in position 3 was studied by …

Suggestion: Reactivity of indole derivatives with iodine at position 3 was studied by …

Abstract: A study of the reactivity of iodine in position 3 was studied by the use of cross-coupling reaction in particular.

Suggestion: A study of the reactivity of iodine in position 3 was studied by the use of cross-coupling reactions.

Abstract: Reaction scheme

Suggestion: Reaction scheme is not shown in the abstract.

Line 29: 2-Cyanoindoles are an important research objective …

Suggestion: 2-Cyanoindoles are important research objectives … or (2-Cyanoindole is an important research objective …)

Figure 1: Examples of biologically relevant compounds featuring 2-cyanoindoles-derived cores.

Suggestion: Biologically relevant compounds featuring 2-cyanoindole cores.

Scheme 3. 30 mn

Suggestion: 30 min.

Line 104: A new serie of indole-2-carbonitrile …

Suggestion: A new series of indole-2-carbonitrile …

Scheme 6: EtOH/eau

Suggestion: What is “eau”?

Line 182: … different alkynes, alcenes and arylics derivatives.

Suggestion: … different alkynes, alkenes, and aryl derivatives.

Line 186: … Sonogashira, Suzuki-Miyaura, Heck and Stille cross-coupling reaction.

Suggestion: … Sonogashira, Suzuki-Miyaura, Heck, and Stille cross-coupling reactions.

Conclusion: Structures

Suggestion: Structures are not shown in the conclusion.

Line 196: … were referenced the central peak of CDCl3 at 77.0 ppm …

Suggestion: … were referenced to the central peak of CDCl3 at 77.0 ppm …

Line 201: … dd, doublet of doublet; dq, doublet of quartet; m, multiplet.

Suggestion: … dd, doublet of doublets; dq, doublet of quartets; m, multiplet.

Line 204: Compounds were visualized with UV light …

Suggestion: Compounds were visualized under UV light …

Line 221: The combined organic layers were dried over …

Suggestion: The combined organic layers was dried over …

Author Response

First of all, we would like to express our gratitude to the referees for their careful examination of our work. We have considered the reviewer’s comments on our manuscript carefully and have made some changes to address the comments of reviewer’s where appropriate. We will now give a detailed response below:

Reviewer 2

The manuscript “Synthesis of New and Highly Functionalized 1H-indole-2-carbonitriles via Cross-Coupling Reactions” describes the syntheses of indole derivatives using cross-coupling reactions.

Biological assays were not performed for the compounds synthesized. The manuscript deserves publication after revision. Listed below are some suggestions for the authors to consider.

Thank you very much for your careful examination of our work and suggestions. Based on your comments, we deeply improved our revised manuscript. The answers to your comments and changes made are listed below.

General comment: It is not indicated where introduction ends, and results and discussions start. After that “Conclusion” is shown. Clear separation of the sections may be indicated.

We used the Microsoft Word Template of Molecules to make the sections appear (Introduction, Results and discussion, Materials and Methods, Conclusions).

 Abstract: A study of the reactivity of iodine in position 3 was studied by …

Suggestion: Reactivity of indole derivatives with iodine at position 3 was studied by …

 Revision effected.

Abstract: A study of the reactivity of iodine in position 3 was studied by the use of cross-coupling reaction in particular.

Suggestion: A study of the reactivity of iodine in position 3 was studied by the use of cross-coupling reactions.

 Revision effected.

 Abstract: Reaction scheme

Suggestion: Reaction scheme is not shown in the abstract.

 The scheme of the abstract has been removed.

Line 29: 2-Cyanoindoles are an important research objective …

Suggestion: 2-Cyanoindoles are important research objectives … or (2-Cyanoindole is an important research objective …)

 Revision effected.

 Figure 1: Examples of biologically relevant compounds featuring 2-cyanoindoles-derived cores.

Suggestion: Biologically relevant compounds featuring 2-cyanoindole cores.

 Revision effected.

Scheme 3. 30 mn

Suggestion: 30 min.

 Revision effected.

Line 104: A new serie of indole-2-carbonitrile …

Suggestion: A new series of indole-2-carbonitrile …

 Revision effected.

Scheme 6: EtOH/eau

Suggestion: What is “eau”?

The conditions of the reaction has been changed into :  Scheme 6. Reagents and conditions: (i) 7a-d (1 equiv.), boronic acids (1.2 equiv.), sat. aq. NaHCO3, EtOH/toluene = 3/2, Pd(PPh3)4 10 mol%, 130 °C, 4 h.

Line 182: … different alkynes, alcenes and arylics derivatives.

Suggestion: … different alkynes, alkenes, and aryl derivatives.

 Revision effected.

Line 186: … Sonogashira, Suzuki-Miyaura, Heck and Stille cross-coupling reaction.

Suggestion: … Sonogashira, Suzuki-Miyaura, Heck, and Stille cross-coupling reactions.

 Revision effected.

Conclusion: Structures

Suggestion: Structures are not shown in the conclusion.

The scheme of the conclusion has been removed.

Line 196: … were referenced the central peak of CDCl3 at 77.0 ppm …

Suggestion: … were referenced to the central peak of CDCl3 at 77.0 ppm …

 Revision effected.

Line 201: … dd, doublet of doublet; dq, doublet of quartet; m, multiplet.

Suggestion: … dd, doublet of doublets; dq, doublet of quartets; m, multiplet.

 Revision effected.

Line 204: Compounds were visualized with UV light …

 Revision effected.

Line 221: The combined organic layers were dried over …

Suggestion: The combined organic layers was dried over …

 The sentence was changed into :The organic layers were combined, dried over MgSO4 and concentrated under reduced pressure.

Please note that all corrections and changes made have been highlighted in yellow in this revision.

Reviewer 3 Report

Dear Authors,

Please note the following issues, which must be addressed before the manuscript can be accepted:

1) Abstract:

Please make it briefer and more informative, avoiding talking of "buliding blocks", active molecules, etc., as they belong to the Introduction. Perhaps a reference to the results obtained...

2) Introduction:

Please, revise references, and take in mind that 2-cyanoindole and indole-2-carbonitrile are synonims! The first paragraph is very redundant.

Refs. 1-3 are not on the preparation of 2-cyanoindoles, but rather on the use of 2-cyanoindoles to elaborate more complex systems.

What is the meaning of "versatile synthetic handles".

3) On Figure 1, reference 11 relates to compound A, ref 12 to B (please note it should be a prostaglandin 2 modulator), and ref 13 to C. However, their carbon frameworks are unralated to the compounds prepared in this manuscript, so some elaboration is needed.

Also, in Figure 1 (line 62) beta-stereochemistries are shown in the forumulas, but there is no chiral center present. Please correct.

4) I think lines 64-65 are a brief abstract of the authors' intentions. However, it is important to find what actual targets of interest can be prepared with the structures shown in the rest of the manuscript.

5) Lines 66-67: "new molecules with biological interest" please add structures related to the compounds made, and the corresponding references.

6) Line 72:

Compound 3a is comercially available, and 3b-d were previously prepared (Organic Letters (2017), 19(19), 5058).  There are several previous preparations of compounds 1a-d, and they might be comercial, too.

The functional group conversion shown in Scheme 1  is unsurprising, but no more details are given in the Experimental part.

7) Scheme 2 (and follows):

Compounds 5, 8, 9, 10 and 11 show a "bold' bond, which is used to indicate a beta stereochemistry. However, there is no such thing in any of the compounds. Please, correct.

8) Line 104:

There is no way to have a triple bond in position 3 in an indole system. Please correct! Perhaps you can have an alkynyl substituent, instead.

9) Line 123 and follows:

The addition of an alkyl lithium to a nitrile is a well known method for the preparation of ketones, and it is the usual product in such reaction. Such lack of functional group tolerance is a known problem with Kumada-Corriú reactions since 1972, and explain the profusion of alternative methods, including mainly Negishi couplings, based in the use of less-reactive nucleophiles.

10) Lines 182-183

Please correct to "alkenes" and "aryl derivatives".

11) Supplementary Information:

Please note that the 13C for  compound 3b is missing.

Also, note that the 13C spectra for compounds 3a, 7a, 8c, 8f, 8g, 9c, 9f, 9g and 9h have problems, as they show "peaks with negative phase", which is impossible for such spectra.

Compound 6c is contaminated with grease, visible in the 13C (at about 30 ppm) and 1H (at about 1.25 ppm). Also, several spectra show presence of solvent, probably hexane, or similar (signals at 1.25 (s) and 0.8 (t) ppm in the 1H spectra).

English needs to be checked thoroughly, as there are several paragraphs that can be easily improved.

The corrections are small, but there are many to make, so I am recommending to reconsider after a major revision.

Author Response

First of all, we would like to express our gratitude to the referees for their careful examination of our work. We have considered the reviewer’s comments on our manuscript carefully and have made some changes to address the comments of reviewer’s where appropriate. We will now give a detailed response below:

Reviewer 3

Dear Authors,

Please note the following issues, which must be addressed before the manuscript can be accepted:

Thank you very much for your careful examination of our work and suggestions. Based on your comments, we deeply improved our revised manuscript. The answers to your comments and changes made are listed below.

1) Abstract:

Please make it briefer and more informative, avoiding talking of "buliding blocks", active molecules, etc., as they belong to the Introduction. Perhaps a reference to the results obtained...

The abstract was changed into : An approach for the preparation of polysubstituted indole-2-carbonitriles through a cross coupling reaction of compounds 1-(but-2-ynyl)-1H-indole-2-carbonitriles and 1-benzyl-3-iodo-1H-indole-2-carbonitriles is described. The reactivity of indole derivatives with iodine at position 3 was studied using of cross-coupling reactions. The Sonogashira, Suzuki-Miyaura, Stille, and Heck cross-couplings afforded a variety of di-, tri- and tetra-substituted indole-2-carbonitriles.

2) Introduction:

Please, revise references, and take in mind that 2-cyanoindole and indole-2-carbonitrile are synonims! The first paragraph is very redundant.

Refs. 1-3 are not on the preparation of 2-cyanoindoles, but rather on the use of 2-cyanoindoles to elaborate more complex systems.

What is the meaning of "versatile synthetic handles".

The introduction has been reworded, examples of compounds of biological interest have been added in Figure 1. In addition, the references from 1 to 9 have been added.

Indole skeletons exist as key building blocks in drugs, natural products, pharmaceuticals, alkaloids, agrochemicals and exhibit potent and wide-ranging biological activities [1-5]. The indole scaffold probably represents one of the most important structural subunits for the discovery of new drug candidates [6-9]. In particular, the derivatives of 2-cyanoindoles gained considerable attention in recent years because of their great importance in biological sciences and they are also of interest thanks to this nitrile function [1012]. 2-Cyanoindole unit is an example of structural motif building blocks and effective precursors for the synthesis of various indole-fused polycycles [1319], substituted 2-cyanoindoles [2024], addition to nitriles [25,26] and indole heterocycle substitution [27,28]. These compounds exhibit a wide range of biological activities (Figure 1). They are widely used in medicinal chemistry and pharmacological research as antagonist molecules. For examples, adrenergic antagonist A [14] is a drug that inhibits the function of adrenergic receptors. There are also α-adreno receptors that are located on vascular smooth muscle. Antagonists reduce or block the signals of agonists. They can be drugs, which are added to the body for therapeutic reasons, or endogenous ligands. Analogue D [27] of firefly luciferin Firefly luciferin is a compound of the class of luciferins, light-emitting chemical compounds. It is found in many species of fireflies (Lampyridae). It is the substrate of luciferase an enzyme that catalyzes its oxidation into oxyluciferin with concomitant hydrolysis of a molecule of ATP into AMP and PPi accompanied by the emission of a photon of yellow light characteristic of these insects. NMDA receptor antagonists E [25] are a class of drugs that work to antagonize, or inhibit the action of, the N-Methyl-D-aspartate receptor (NMDAR). They are commonly used as anesthetics for animals and humans; the state of anesthesia they induce is referred to as dissociative anesthesia. The dopamine D4 receptor (D4R) F [20] plays important roles in cognition, attention, and decision making. Novel D4R-selective ligands have promise in medication development for neuropsychiatric conditions, including Alzheimer’s disease and substance use disorders. Prostaglandin E2 (PGE2) modulator G [21], subtype (EP2), which is a metabolite of arachidonic acid that binds with and regulates cellular responses to PGE2, is associated with numerous physiological and pathological events in a wide range of tissues. As a stimulatory G protein‑coupled receptor, PGE2‑induced EP2 activation can activate adenylate cyclase, leading to increased cytoplasmic cAMP levels and activation of protein kinase A. Finally, compound H [22] is considered as antiarrhythmic agent. Also, the cyano group is a valuable and readily available functional group for the preparation of various functional groups such as amines, amides, esters, ketones, and their carboxyl derivatives [29].

3) On Figure 1, reference 11 relates to compound A, ref 12 to B (please note it should be a prostaglandin 2 modulator :  Revision effected), and ref 13 to C. However, their carbon frameworks are unralated to the compounds prepared in this manuscript, so some elaboration is needed.

Also, in Figure 1 (line 62) beta-stereochemistries are shown in the forumulas, but there is no chiral center present. Please correct.

 Revision effected.

4) I think lines 64-65 are a brief abstract of the authors' intentions. However, it is important to find what actual targets of interest can be prepared with the structures shown in the rest of the manuscript.

These molecules are currently being tested through a collaboration with the N2C team at University of Tours on calcium channels in breast cancer. These experiments are currently underway.

This will be the subject of another publication at the chemistry-biology interface.

5) Lines 66-67: "new molecules with biological interest" please add structures related to the compounds made, and the corresponding references.

Examples of molecules of biological interest have been added in the introduction and figure 1.

6) Line 72:

Compound 3a is comercially available, and 3b-d were previously prepared (Organic Letters (2017), 19(19), 5058).  There are several previous preparations of compounds 1a-d, and they might be comercial, too.

In the experimental part, the references of preparation of products 3a-d appear well.  These products are not described in the cited reference (OL 2017, 19, 5058) it is a cyanation method (reference 35 in the manuscript).

They are very expensive products (1H-indole-2-carbonitrile : 1 g = 223.60 € Merck ; 5-methoxy-1H-indole-2-carbonitrile : starting at 1290 $ Aurora Fine Chemicals…  , so we preferred to synthesize them.

The functional group conversion shown in Scheme 1  is unsurprising, but no more details are given in the Experimental part.

The first steps are well described in the literature (synthesis of acid chlorides, synthesis of amides), we preferred to put in the experimental part only the last step of preparation of the nitrile function from amides.

7) Scheme 2 (and follows):

Compounds 5, 8, 9, 10 and 11 show a "bold' bond, which is used to indicate a beta stereochemistry. However, there is no such thing in any of the compounds. Please, correct.

Revision effected for all schemes.

8) Line 104:

There is no way to have a triple bond in position 3 in an indole system. Please correct! Perhaps you can have an alkynyl substituent, instead.

Revision effected.

9) Line 123 and follows:

The addition of an alkyl lithium to a nitrile is a well known method for the preparation of ketones, and it is the usual product in such reaction. Such lack of functional group tolerance is a known problem with Kumada-Corriú reactions since 1972, and explain the profusion of alternative methods, including mainly Negishi couplings, based in the use of less-reactive nucleophiles.

Other tests with organomagnesiums have been tested. The formation of the ketone is observed each time.

10) Lines 182-183

Please correct to "alkenes" and "aryl derivatives".

 Revision effected.

11) Supplementary Information:

Please note that the 13C for  compound 3b is missing.

The 13C NMR spectrum for compound 3b whas been added.

Also, note that the 13C spectra for compounds 3a, 7a, 8c, 8f, 8g, 9c, 9f, 9g and 9h have problems, as they show "peaks with negative phase", which is impossible for such spectra.

In fact, these are J-mod experiments. This has been corrected in the SI (page 1).

Compound 6c is contaminated with grease, visible in the 13C (at about 30 ppm) and 1H (at about 1.25 ppm). Also, several spectra show presence of solvent, probably hexane, or similar (signals at 1.25 (s) and 0.8 (t) ppm in the 1H spectra).

On some spectra, we have annotated the presence of grease.

English needs to be checked thoroughly, as there are several paragraphs that can be easily improved.

English has been carefully proofread, typographicals and gramaticals errors have been corrected.

The corrections are small, but there are many to make, so I am recommending to reconsider after a major revision.

Please note that all corrections and changes made have been highlighted in yellow in this revision.

Round 2

Reviewer 2 Report

The manuscript “Synthesis of New Highly Functionalized 1H-indole-2-carbonitriles via Cross-Coupling Reactions” describes the syntheses of indole derivatives using cross-coupling reactions.

Biological assays were not performed for the compounds synthesized. Authors have implemented the suggested corrections. The manuscript deserves publication after minor revision.

Title of the manuscript: Synthesis of New Highly Functionalized 1H-indole-2-carbonitriles via Cross-Coupling Reactions Title

Suggestion: Synthesis of New Highly Functionalized 1H-indole-2-carboni- 2

triles via Cross-Coupling Reactions

The extra word “Title” is not part of the “Title of the manuscript”

Line 770: The present work allows an access to …

Suggestion: The present work allows access to …

Author Response

Title of the manuscript: Synthesis of New Highly Functionalized 1H-indole-2-carbonitriles via Cross-Coupling Reactions Title

Suggestion: Synthesis of New Highly Functionalized 1H-indole-2-carboni- 2

triles via Cross-Coupling Reactions

The extra word “Title” is not part of the “Title of the manuscript”

Revision effected.

Line 770: The present work allows an access to …

Suggestion: The present work allows access to …

Revision effected.

Reviewer 3 Report

Dear Authors,

All issues have appropriately been addressed, and the manuscript can be accepted after a small correction:

I think that if the 13C spectra were obtained using a special technique, such as DEPTQ, a short note should be added in the Experimental part, and explicit how these spectra were made.

Best regards,

Author Response

I think that if the 13C spectra were obtained using a special technique, such as DEPTQ, a short note should be added in the Experimental part, and explicit how these spectra were made.

In the "Materials and methods" (General Information), the sentence :

"1H NMR spectra were recorded in CDCl3 or referenced the residual CHCl3 at 7.26 ppm (2.50 ppm for DMSO-d6), and 13C NMR spectra were referenced to the central peak of CDCl3 at 77.0 ppm (39.52 ppm for DMSO-d6)" was changed into:

"1H NMR spectra were recorded in CDCl3 or referenced the residual CHCl3 at 7.26 ppm (2.50 ppm for DMSO-d6), 13C NMR and J-mod spectra were referenced to the central peak of CDCl3 at 77.0 ppm (39.52 ppm for DMSO-d6)".

It is not necessary to specify this sequence, because we all know that a J-mod spectrum allows to distinguish quaternary carbons and CH2 at the bottom and CH and CH3 at the top.